

# Retrieving Vertical Ozone Profiles from Measurements of Global Spectral Irradiance

Germar Bernhard[1], Irina Petropavlovskikh[2,3], Bernhard Mayer[4]

[1]Biospherical Instruments Inc., San Diego, CA 92110, USA
[2]Cooperative Institute for Research in Environmental Sciences (CIRES), University of Colorado, Boulder, CO 80309, USA
[3]NOAA Earth System Research Laboratory, Global Monitoring Division, Boulder, CO 80305, USA
[4]Ludwig-Maximilians-Universität, Munich, 80333, Germany

*Correspondence to*: Germar Bernhard (bernhard@biospherical.com)

**Abstract.** A new method is presented to determine vertical ozone profiles from measurements of spectral global (direct Sun plus upper hemisphere) irradiance in the UV. The method is similar to the widely used Umkehr technique, which inverts measurements of zenith sky radiance. The procedure was applied to measurements of a high-resolution spectroradiometer installed near the centre of the Greenland ice sheet. Retrieved profiles were validated with balloon sonde observations and ozone profiles from the space-borne Microwave Limb Sounder (MLS). Depending on altitude, the bias between retrieval results presented in this paper and MLS observations ranges between −5 % and +3 %. The magnitude of this bias is comparable, if not smaller, to values reported in the literature for the standard Dobson Umkehr method. Total ozone columns (TOCs) calculated from the retrieved profiles agree to within 0.7±2.0 % (±1σ) with TOCs measured by the Ozone Monitoring Instrument (OMI) onboard the Aura satellite. The new method is called the "Global-Umkehr" method.

## 1 Introduction

The "Umkehr" method for determining the vertical distribution of ozone in the atmosphere was first introduced in the 1930s (Götz et al., 1934) and is now routinely applied to measurements of Dobson (e.g, Dütsch, 1959, Mateer and DeLuisi, 1992; Petropavlovskikh et al, 2005) and Brewer (McElroy and Kerr, 1995; Petropavlovskikh et al., 2011) spectrophotometers. The method is typically based on analyzing ratios of zenith-sky radiances at two wavelengths in the ultraviolet (UV), one strongly and one weakly attenuated by ozone, that are measured at solar zenith angles (SZAs) between 60° and 90°. Here we explore a similar optimal statistical approach to obtain vertical ozone information from measurements of spectrally resolved global irradiance, i.e., the irradiance received by a horizontal "cosine" collector from direct Sun and sky (upper hemisphere, from zenith to horizon). Such measurements are routinely performed by several UV monitoring networks sponsored by the NSF (http://uv.biospherical.com/), NOAA (http://www.esrl.noaa.gov/gmd/grad/antuv/), the Network for the Detection of Atmospheric Composition Change (NDACC; http://www.ndsc.ncep.noaa.gov/), Environment Canada (http://exp-studies.tor.ec.gc.ca/e/ozone/ozonecanada.htm), and others. The proposed method has the potential to make these long-term



data sets available for assessing vertical ozone information in an approach similar to Umkehr retrievals. This is particularly
interesting for locations where zenith-sky observations are not available.
Compared to other methods (e.g., Lidar observations (Megie et al., 1977), balloon sondes, and microwave spectrometers
(Parrish et al., 1992; Waters et al., 2006)), the Umkehr technique provides a relatively inexpensive way of measuring the
vertical distribution of ozone in the atmosphere. The method is most sensitive to the altitude range of 20 to 45 km and has a
resolution of about 10 km within this range. For mid-latitude sites, Brewer Umkehr data have a precision of about 15 % in
the 20- to 40-km region, with larger departures outside this altitude range (McElroy and Kerr, 1995). Umkehr data are
routinely used for monitoring the drift of sensors measuring the vertical distribution from ozone from space (Newchurch et
al. 1987; DeLuisi et al, 1994; Miller et al. 1997; Krzyścin  et al, 2009; Petropavlovskikh et al., 2005; Petropavlovskikh et al,

10  2011).

The use of measurements of global irradiance instead of zenith-sky radiance for Umkehr retrievals is of no advantage *per se*.
First, global irradiance includes the direct solar beam, which is attenuated according to Beer's law and therefore does not
contain information on the profile. Second, global irradiance includes photons received from directions close to the horizon
and multiple-scattering effects are therefore not negligible. We will show that both challenges can be overcome, resulting in
profiles of similar accuracy than those inverted from zenith-sky observations. The main advantage of the method presented
here is that the vertical distribution of ozone can be derived for locations where no other ground-based data exist from which
profiles could be calculated. The new method is called the "Global-Umkehr" method.
The Global-Umkehr method was tested using data from the NSF UV Monitoring Network (Booth et al., 1994), which has
been measuring UV and visible global spectral irradiance (290 – 600 nm) at six high-latitude sites since 1990. For this study
we used data from Summit, Greenland (72° 35' N, 38° 27' W, 3,202 m a.s.l) where ozone profiles have been routinely
measured also by balloon sondes. The method can also be applied to measurements at lower latitude sites.

## 1 Method

### 1.1 Retrieval method

The retrieval method is based on the optimal estimation approach (Gauss-Newton method) developed by Rodgers (2000). In
brief, the solution (i.e., the ozone concentration as a function of altitude or pressure) is determined iteratively with the matrix
equation:
$$\mathbf{x}_{i+1} = \mathbf{x}_i + \mathbf{S}_{i+1}[\mathbf{K}_i^T \mathbf{S}_\varepsilon^{-1}(\mathbf{y} - \mathbf{F}(\mathbf{x}_i)) - \mathbf{S}_a^{-1}(\mathbf{x}_i - \mathbf{x}_a)] \tag{1}$$
where
$$\mathbf{S}_{i+1} = (\mathbf{S}_a^{-1} + \mathbf{K}_i^T \mathbf{S}_\varepsilon^{-1}\mathbf{K}_i)^{-1}. \tag{2}$$
Eqs. (1) and (2) contain the following parameters:





$\mathbf{x}_i$     is the "state vector" of iteration $i$. In our implementation, it is defined as the average ozone concentration in eleven layers with a layer thickness of 5 km.

$\mathbf{y}$     is the "measurement vector," which is composed of ratios of global spectral irradiance $E(\lambda)$ measured at 310 nm (a wavelength strongly attenuated by ozone) and 337 nm (a wavelength weakly attenuated by ozone) for SZAs ranging between 70° and 90°.

$\mathbf{F}(\mathbf{x}_i)$     is the solution of the forward model (Sect. 2.3), which simulates the measurements using the state vector as input.

$\mathbf{K}_i$     is the Jacobian matrix of the partial derivatives of the forward model results and the state vector.

$\mathbf{S}_\varepsilon$     is the covariance matrix quantifying the uncertainty of the measurements.

$\mathbf{x}_a$     is the *a priori* state vector. The iteration starts by setting $\mathbf{x}_0 = \mathbf{x}_a$.

$\mathbf{S}_a$     is the covariance matrix pertaining to the *a priori* state vector.

$\mathbf{S}_{i+1}$     is the solution error covariance matrix at iteration $i+1$, which can be exploited to calculate the uncertainty of the retrieval.

We chose 310 nm as the lower wavelength because measurements at this wavelength are at least a factor of 50 larger than
the spectroradiometer's detection limit of 0.001 mW m$^{-2}$ nm$^{-1}$ for all SZAs and ozone columns of interest. The upper
wavelength of 337 nm was chosen because the temperature sensitivity of the ozone absorption cross section has a local
minimum at about this wavelength (Bass and Paur, 1985). We also tested other wavelength pairs or combinations of several
pairs of wavelengths (e.g., combinations of $E(305)/E(337)$; $E(310)/E(337)$; $E(325)/E(337)$) when developing the method. We
found that the use of multiple pairs improved the information content only minimally but increased the computational time
considerably.
The SZA range chosen for Umkehr observation is a trade-off between the additional information content resulting from a
larger range and the risk that environmental conditions (e.g., clouds, ozone profile) may change substantially over the longer
observation time that a larger SZA range requires. During development, we tried several SZA ranges and found that a range
of 70° to 90° is a good compromise. This observation is consistent with the conclusion of Petropavlovskikh et al. (2005) that
information in the upper layers is not degraded by changing the SZA range from 60°–90° to 70°–90° in the standard Umkehr
method. We also omitted observations with SZAs larger than 90° because of potential systematic errors in the forward model
results (Sect. 2.3) when the Sun is below the horizon. At the latitude of Summit, a SZA range of 70° to 90° is available in
spring between 27 March and 8 May and in fall between 4 August and 15 September.
The Jacobian matrix $\mathbf{K}_i$ has the elements $[\mathbf{K}_i]_{mn} = [\partial \mathbf{F}(\mathbf{x}_i)]_m / [\partial \mathbf{x}_i]_n$ and is calculated for every iteration step.
The measurement error covariance matrix $\mathbf{S}_\varepsilon$ is a diagonal matrix and is constructed by assuming that elements of the
measurement vector have an uncertainty of $\sigma_\varepsilon = 3\%$ and are independent of wavelength and SZA:

$$[\mathbf{S}_\varepsilon]_{mn} = \begin{cases} \sigma_\varepsilon^2 [\mathbf{y}]_m [\mathbf{y}]_n & \text{for } m = n \\ 0 & \text{for } m \neq n \end{cases} \tag{3}$$



The value of 3 % was chosen based on the uncertainty budget of the spectroradiometer installed at Summit (Sect. 2.2). The
choice of 3 % was further supported by analyzing the residuals of the retrieval results ( $\mathbf{y} - \mathbf{F}(\hat{\mathbf{x}})$ ) where $\hat{\mathbf{x}}$ indicates the
solution state vector after the final iteration.
*A priori* state vectors $\mathbf{x}_a$ were constructed by combining balloon sonde profiles for altitudes below 10 km and profiles
measured by the Microwave Limb Sounder (MLS) on NASA's Aura satellite for altitudes above 10 km (see Sect. 2.5 for
additional information on these profiles). Separate *a priori* profiles were used for processing data from spring (27 March – 8
May) and fall (4 August – 15 September). Profiles for both seasons were constructed by calculating the median of a large
number of sonde and MLS profiles measured during the two periods using data from the years 2004 to 2014.
The covariance matrix pertaining to the *a priori* state vector, $\mathbf{S}_a$ , was constructed as suggested by Bhartia et al. (2013):
$$[\mathbf{S}_a]_{mn} = \sigma_a^2 [\mathbf{x}_a]_m [\mathbf{x}_a]_n \exp(-|m-n|/d) . \tag{4}$$
The parameter $\sigma_a$ specifies the anticipated variability of the retrieved profiles about the *a priori* profile and can be
interpreted as the relative standard deviation of the profiles' distribution. The correlation length $d$ was set to two, which is
equivalent to 10 km for our definition of the state vector.
When $\sigma_a$ is set to a small value (e.g., 0.1), the solution of the inversion becomes very sensitive to the *a priori* profile. In
contrast, when $\sigma_a$ is set to a large value, the solution is mostly determined by the measurements. Choosing the optimum
value for $\sigma_a$ is a trade-off between two competing effects: a large value of $\sigma_a$ ensures correct inversion result even if the
true profile deviates greatly from the *a priori* profile. On the other hand, a small value of $\sigma_a$ reduces the risk that the
retrieval result is grossly incorrect if measurements are affected by unanticipated errors.
We calculated profiles for $\sigma_a = 0.1$ and 0.4, and compare the results in Sect. 3. The value of $\sigma_a = 0.1$ was chosen by
analyzing the variability of MLS profiles relative to the spring and fall *a priori* profiles introduced above. For Umkehr layers
3 though 7 (the layers for which the Umkehr method is most sensitive) the relative standard deviations calculated from the
MLS profiles vary between 0.05 and 0.15; averaged over layers 3 though 7, the relative standard deviation is 0.12 for the
spring and 0.09 for the fall period. The value of $\sigma_a = 0.4$ was chosen as the other extreme. With this value, the *a priori*
profile has little influence on the inversion result and the effect of errors in the measurement vector $\mathbf{y}$ becomes more
prominent. Of note, the retrieval results depends technically on the ratio $\gamma \equiv (\sigma_\varepsilon / \sigma_a)^2$ as opposed to $\sigma_a$ (Bhartia et al.,
2013). Because the measurement uncertainty $\sigma_\varepsilon$ is well defined, we discuss the results using $\sigma_a$ instead of $\gamma$.
The iteration is repeated until two conditions are met: first, the norms of $\mathbf{x}_{i+1}$ and $\mathbf{x}_i$ must differ by less than 0.5 %, and
second, the values of consecutive results of the cost function $\Psi(\mathbf{x})$ must agree to within 5.0 %, where
$$\Psi(\mathbf{x}) = (\mathbf{y} - f(\mathbf{x}))^T \mathbf{S}_\varepsilon^{-1} (\mathbf{y} - f(\mathbf{x})) + (\mathbf{x}_a - \mathbf{x})^T \mathbf{S}_a^{-1} (\mathbf{x}_a - \mathbf{x}) . \tag{5}$$



These convergence criteria were adopted from Tzortziou et al. (2008). We confirmed that these criteria are also appropriate
for our application by analyzing changes of the two convergence metrics as a function of iteration $i$. The two criteria are
always met in two to four iterations.
The uncertainty $e_m$ of each element of the solution's state vector was calculated according to Goering et al. (2005) from the
diagonal elements of the solution error covariance matrix and the solution state vector:

$$e_m = \frac{\sqrt{[\hat{\mathbf{S}}]_{mm}}}{[\hat{\mathbf{x}}]_m},$$
(6)

where the caret (^) above the symbols $\mathbf{x}$ and $\mathbf{S}$ indicates the values of $\mathbf{x}_i$ and $\mathbf{S}_i$ of the final iteration.
The optimal estimation technique provides several diagnostics about the quality of the retrieved profile. The diagnostic used
here is $d_s$, which expresses the "number of degrees of freedom for signal" and indicates the number of useful independent
observations in the measurement vector $\mathbf{y}$. $d_s$ was calculated as suggested by Rodgers (2000) and Goering et al. (2005)
from the singular values $\lambda_m$ of the "error-weighted weighting function matrix" $\widetilde{\mathbf{K}} \equiv \mathbf{S}_\varepsilon^{-1/2} \mathbf{K} \mathbf{S}_a^{-1/2}$ via:

$$d_s = \sum_m \frac{\lambda_m^2}{1 + \lambda_m^2}.$$
(7)

The diagnostic $d_s$ depends on $\mathbf{S}_a$ and in turn on $\sigma_a$. We will show in Sect. 3 that $d_s$ is considerably smaller for profiles
calculated with $\sigma_a = 0.1$ than 0.4.
**2.2 Measurements**
The method was tested using measurements of global spectral irradiance performed at Summit with a SUV-150B
spectroradiometer designed by Biospherical Instruments Inc. The instrument has a spectral resolution of 0.63 nm, is part of
the U.S. National Science Foundation's Arctic Observing Network and contributes data to NDACC. The expanded
uncertainty (coverage factor $k$ = 2, equivalent to uncertainties at the 2σ-level or a confidence interval of 95 %) of global
spectral irradiance measurements for wavelengths between 310 and 337 nm is between 6.0 and 6.7 %. More information on
the instrument is provided by Bernhard et al. (2008) and a detailed uncertainty budget is available at
http://uv.biospherical.com/Version2/Uncertainty_SUV150B.pdf.
Data used in this paper are from the "Version 2" data edition (Bernhard et al., 2004) and are corrected for the cosine error of
the instrument's entrance optics. The wavelength mapping was determined with a Fraunhofer-line correlation method and
the wavelength uncertainty ($k$ = 2) of processed data is 0.02 nm. Measured spectra and spectra calculated with the forward
model (Sect. 2.3) were convolved with a triangular function of 2 nm bandwidth to further reduce uncertainties resulting from
potential wavelengths shifts between measured and modelled spectra.





The SUV-150B is a scanning instrument, which measures each wavelength at a different time. The time required to scan
between 310 and 340 nm is about 140 seconds. Changing cloud condition will therefore affect the ratio of measurements at
these wavelengths, and in turn the accuracy of the retrieval result. The effect of clouds on the ratio of $E(310)/E(337)$ can be
reduced using measurements of a filtered photodiode, which is illuminated via a beam splitter located between the entrance
optics and monochromator of the SUV-150B system. The sensitivity of the diode is centred at 330 nm and measurements are
preformed continuously during the recording of spectra. Because attenuation of thin clouds is fairly uniform in the 310 to
337 nm range (Seckmeyer et al., 1996), measurements of the photodiode can be used to correct for variable cloud
attenuation. Specifically, spectral measurements at $\lambda = 310$ nm or $\lambda = 337$ nm are multiplied with a correction factor
$C(\lambda,t)$, defined as:
$$C(\lambda,t) = \frac{D_C(\theta(t))}{D(t)},\qquad\qquad(8)$$
where $t$ is the time of the spectral measurement, $\theta(t)$ is the SZA at time $t$, $D(t)$ is the measurement of the photodiode at
time $t$, and $D_C(\theta(t))$ is the measurement of the photodiode during clear skies at $\theta(t)$. $D_C(\theta(t))$ was parameterized as a
function of SZA by filtering measurements for clear skies. The correction takes into account that the SZA changes between
measurements at 310 and 337 nm. Of note, this technique cannot be applied in the presence of optical thick clouds which
enhance ozone absorption of tropospheric ozone due to path length enhancement (Mayer et al., 1998). This restriction does
not apply to Summit where clouds are always optically thin (Bernhard et al., 2008).
Measurement vectors were inverted both with and without the cloud correction and results are compared in Sect. 3.2.
Spectral irradiances at 310 and 337 nm were calculated for all spectra measured during a given period of Umkehr
observations and interpolated to a common SZA grid (70, 75, 80, 85, 87, 88, 89, and 90°) using an approximating
(smoothing) spline. Compared to an interpolating spline, an approximating spline has the advantage to reduce noise in the
measurement vector further.
The measurement vector is only constructed from spectra measured in the afternoon (between 15:00 and 20:00 UTC)
because solar measurements have gaps in the morning when the system performs diagnostics scans with internal lamps
(wavelength and irradiance standards).
**2.3 Forward model**
Forward modelling was performed with Version 1.01 of the pseudospherical discrete ordinate (SDISORT) radiative transfer
solver of the UVSPEC/libRadtran model (Mayer and Kylling, 2005). The number of streams was set to 12. The model's
result are spectra of global irradiance. Model input parameters include the extraterrestrial spectrum as defined by Bernhard et
al. (2004) and available at http://uv.biospherical.com/Version2/Paper/2004JD004937-ETS_GUEYMARD.txt; surface
albedo; the ozone absorption cross section (Bass and Paur, 1985); and atmospheric pressure. The surface albedo at Summit
was set to 0.97 in good agreement with recent measurements (Carmagnola et al., 2013). Aerosol optical depth was set to



stratospheric background conditions. Atmospheric pressure and profiles of gases other than ozone ($O_2$, $H_2O$, $CO_2$, and $NO_2$)
were taken from the AFGL atmospheric constituent profile for subarctic summer (Anderson et al., 1986), which defines the
atmosphere at 51 levels. The vertical distribution of ozone in this standard profile was replaced with the profile defined by
the state vector $\mathbf{x}_i$ and updated in every iteration.
The SDISORT solver has been successfully validated using data of the NSF UV Monitoring Network (e.g., Bernhard et al.,
2004, 2008) and for a large range of conditions at other sites (e.g., Mayer and Kylling, 2005, and reference therein).
However to the best of our knowledge, a rigorous validation for the large SZAs required for Umkehr retrievals has not been
conducted. The pseudospherical approximation used by SDISORT correctly describes the attenuation of the direct beam in
spherical geometry but the diffuse radiance is computed in plane-parallel geometry (Mayer et al., 2015). This approximation
can lead to significant errors at large SZAs (Petropavlovskikh et al., 2000; Emde and Mayer, 2007). To quantify these errors
for our application, we have compared spectra of global irradiance calculated with SDISORT with the spherical solver of the
MYSTIC (Monte Carlo code for the phYSically correct Tracing of photons In Cloudy atmospheres) model, which fully
solves the spherical geometry without any approximations (Mayer, 2009). Both models were run with the same set of input
parameters (AFGL subarctic summer with *a priori* ozone profile for spring at Summit) for wavelengths between 307 and
313 nm and between 334 and 340 nm in 0.5 nm steps. The MYSTIC model was run with 84 million photons per wavelength
and per SZA, resulting in photon noise of less than 0.5 % at SZA=90° (worst case). Resulting spectra of both models were
convolved with a triangular function of 2 nm bandwidth to further reduce noise and to be consistent with the method used in
the Umkehr code.
Fig. 1a shows the ratio of SDISORT and MYSTIC spectra calculated for the eight SZAs used in our Umkehr
implementation. SDISORT overestimates spectral irradiances relative to MYSTIC at all wavelengths and SZAs. For SZA
$\leq 88°$, the bias is less than 2 % but increases to up to 6.5 % for SZA = 90°. For the Umkehr retrieval, only the ratio
$q(\theta) \equiv \dfrac{E(310,\theta)}{E(337,\theta)}$ is important where $\theta$ indicates again the SZA. The ratio $R(\theta) \equiv \dfrac{q_{SDISORT}(\theta)}{q_{MYSTIC}(\theta)}$ resulting from calculating
$q(\theta)$ with SDISORT and MYSTIC is shown in Fig. 1b. $R(\theta)$ ranges between 0.998 at 80° to 1.019 at 90°. Calculations
with the MYSTIC model can be considered the most accurate results attainable because the Monte Carlo code does not use
approximations. The model has been validated by comparison to other spherical radiative transfer models and by simulating
the radiance distribution of the sky during a total solar eclipse. For such calculations, a spherical solver without
approximations is required because light entering the atmosphere more than 1000 km away may impact the radiance in the
centre of the umbral shadow (Emde and Mayer, 2007).





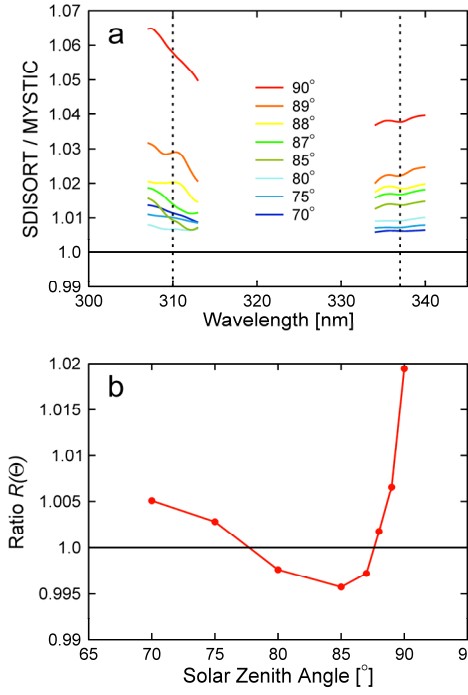

Fig. 1. Comparison of results calculated with the SDISORT and MYSTIC models. (a) Ratio of SDISORT and MYSTIC

spectra calculated for eight SZAs (see legend). (b) Ratio $R(\theta)$. See text for definition.

Relative to MYSTIC, SDISORT overestimates $q(\theta)$ for SZA larger than 88°. In our Umkehr code, we scale the results of

the forward model with $1/R(\theta)$ to account for the bias of the SDISORT model. Note that the MYSTIC model is too slow to

be used for Umkehr retrievals: the calculation of the eight spectra used for Fig. 1a required a run time of over three days.

The forward model requires that the vertical structure of the atmosphere is defined as a function of altitude. The association

between altitude and pressure is defined in the AFGL profile and this relationship may differ from the actual pressure profile

at the time of Umkehr observation. Because our measurements do not allow to reconstruct the pressure profile, we report all

ozone profiles as a function of pressure, and compare the retrieved profile with sonde and MLS profiles, which are also

provided as a function of pressure. The standard Umkehr technique (Petropavlovskikh et al., 2005) uses a similar approach.

Table 1 provides altitude and pressure ranges for each Umkehr Layer. Note that Layer 0 starts at the elevation of Summit

(3,202 m).





Table 1. Assignment of Umkehr Layers.

| Umkehr Layer | Altitude range forward model [km] | Pressure range [hPa] |
| --- | --- | --- |
| 10 | 50.0 – 55.0 | 0.987 – 0.537 |
| 9 | 45.0 – 50.0 | 1.82 – 0.987 |
| 8 | 40.0 – 45.0 | 3.40 – 1.82 |
| 7 | 35.0 – 40.0 | 6.61 – 3.40 |
| 6 | 30.0 – 35.0 | 13.4 – 6.61 |
| 5 | 25.0 – 30.0 | 27.8 – 13.4 |
| 4 | 20.0 – 25.0 | 59.0 – 27.8 |
| 3 | 15.0 – 20.0 | 126.0 – 59.0 |
| 2 | 10.0 – 15.0 | 267.7 – 126.0 |
| 1 | 5.0 – 10.0 | 541.0 – 267.7 |
| 0 | 3.202 – 5.0 | 664 – 541 |

**2.4 Averaging Kernels**
The performance of an inversion based on the optimal estimation approach is often assessed with the averaging kernel matrix
$\mathbf{A} \equiv \hat{\mathbf{S}}\mathbf{K}_i^T \mathbf{S}_\varepsilon^{-1}\mathbf{K}_i$ , which quantifies the sensitivity of the retrieved state $\hat{\mathbf{x}}$ to perturbations in the true state $\mathbf{x}$ . For an ideal
observing system, $\mathbf{A}$ is the identity matrix. In reality, the rows of the averaging kernel matrix are peaked with a finite width,
which can be regarded as a measure of the vertical resolution of the retrieved profile. Similarity to the identity matrix
indicates that the retrieval solution has been determined using the observations rather than the *a priori* information, and as
such, the retrieval has provided new information about the actual state.
Elements of $\mathbf{A}$ can have large positive and negative values for layers where the ozone concentration is close to zero. To
prevent this predicament, Bhartia et al. (2013) suggested to illustrate the performance of the algorithm with "relative
averaging kernels" (RAK or $\mathbf{A}_R$ ), which quantify the relative change of the retrieved state $\hat{\mathbf{x}}$ to the perturbations in the true
state $\mathbf{x}$ . $\mathbf{A}_R$ is defined by
$$[\mathbf{A}_R]_{mn} = [\mathbf{A}]_{mn} \frac{[\hat{\mathbf{x}}]_n}{[\hat{\mathbf{x}}]_m} . \qquad (9)$$
**2.5 Validation method**
The retrieved Umkehr profiles were validated using ozone profiles measured at Summit with balloon sondes by
NOAA/GMD (Oltmans et al., 2010) and ozone profiles provided by MLS on Aura. Sondes are typically launched between



12:00 and 20:00 UTC.  MLS measure thermal emissions from rotational lines of ozone through the limb of the atmosphere.
Ozone measurements have a vertical range of 12–73 km with a vertical resolution of 2–3 km below 65 km. The horizontal
resolution is about 200 km and the accuracy is about 5–10 % between 16 and 60 km (Froidevaux et al., 2008). Sonde and
MLS profiles were downloaded from ftp://ftp.cmdl.noaa.gov/ozwv/Ozonesonde/Summit,%20Greenland/ and
http://avdc.gsfc.nasa.gov/pub/data/satellite/Aura/MLS/V04/L2GPOVP_Prof/O3/Summit/, respectively. Sonde profiles are
only available for 2 to 4 days per month whereas MLS profiles are available on a daily basis.
The total ozone column (TOC) was calculated from the retrieved Umkehr profiles and compared with measurements of the
Ozone Monitoring Instrument (OMI) on board NASA's Aura spacecraft. OMI overpass data were downloaded from
http://avdc.gsfc.nasa.gov/index.php?site=1593048672&id=28. OMI data use the Bass and Paur (1985) ozone absorption
cross section (pers. comm., David Haffner, NASA), like the forward model.
Good validation results can only be expected if the actual ozone profile does not change over the period of Umkehr
observations. We therefore only considered periods where the TOC measured by OMI did not change by more than 20 DU
between 15:00 UTC on the day of the comparison and the first observation on the following day. Retrieved Umkehr profiles
were compared with the sonde profile measured on the same day (if available) and with the MLS profiles measured on this
day (labelled "MLS 1" in the following) as well as the next day ( labelled "MLS 2").
**3 Results**
We first show retrieval results for three sample days with greatly different conditions and compare these results with profiles
measured by balloon sondes and MLS (Sect. 3.1). We then discuss in Sect. 3.2 statistics for all profiles that were retrieved
under sufficiently stable conditions (variation in total ozone of less than ±20 DU).
**3.1 Comparison with balloon sonde and MLS profiles – sample profiles**
**3.1.1 Validation for 19 April 2014**
Fig. 2 compares the retrieved ozone profile for 19 April 2014 with the *a priori*, balloon sonde and MLS profiles. OMI
measured a TOC of 461 DU on this day, which was the third highest TOC of the dataset and the highest TOC of days when
balloon sonde data were available. The profile represents therefore one of the highest departures from the spring *a priori*
profile.





Fig. 2. Validation of ozone profile retrieved for 19 April 2014. Top row: results for $\sigma_a = 0.4$ and uncorrected forward model. Centre row: results of $\sigma_a = 0.4$ and corrected forward model. Bottom row: $\sigma_a = 0.1$, and corrected forward model. 1st column: ozone concentration as a function of pressure for *a priori* profile (grey), balloon sonde profile (blue), MLS profile for day of retrieval (MLS 1, dark green), MLS profile of the following day (MLS 2, light green), and retrieved profile (red). Solid or open circles indicate for each dataset ozone concentrations averaged over each of the eleven Umkehr layers defined in Table 1. Grey error bars indicate the diagonal elements of $\mathbf{S}_a$. Red error bars indicate the uncertainty of the retrieval $e_m$. TOCs measured by OMI and calculated from the retrieved profile are indicated in the legend. 2nd column: layer ozone as a function of pressure for *a priori* profile, balloon sonde profile, MLS profile for day of retrieval, MLS profile of the following day, and retrieved profile. 3rd column: difference between the retrieval and sonde, MLS 1 and MLS 2 datasets averaged over each Umkehr layer. 4th column: relative averaging kernels.





Results are shown for three sets of retrieval parameters: (1) $\sigma_a$ = 0.4, forward model not corrected (top row of Fig. 2); (2)
$\sigma_a$ = 0.4, forward model corrected by scaling with $1/R(\theta)$ (centre row of Fig. 2); and (3) $\sigma_a$ = 0.1, forward model
corrected (bottom row of Fig. 2). For each set of parameters, we show profiles of ozone concentrations (1st column of Fig.
2), layer ozone (2nd column of Fig. 2), the difference between the retrieved profile and the profiles measured by sondes and
MLS (3rd column of Fig. 2), and the relative averaging kernels (RAKs) of the retrieval (4th column of Fig. 2).
For each dataset shown in Fig. 2a, e, and i, solid or open circles indicate the ozone concentrations averaged over each of the
eleven Umkehr layers defined in Table 1. Layer ozone (Fig. 2b, f, and j) was calculated by integrating these average values
over height. Note that ozone concentrations (Fig. 2a, e, and i) are plotted on a linear scale to highlight differences in the
troposphere and lower stratosphere, while layer ozone (Fig. 2b, f, and j) is plotted on a logarithmic scale to better distinguish
differences in the upper stratosphere.
Fig. 2c, g, and k show differences of the average ozone concentrations for the 11 Umkehr layers. Two MLS datasets are
considered. The dataset labelled "MLS 1" is from the same day as the retrieval while the dataset labelled "MLS 2" is from
the following day.
When plotting ozone concentrations on a linear scale (Fig. 2a, e, i), results for the three sets of parameters look similar. As
expected, the resolution of the retrieval is not sufficient to capture the large fluctuation in the ozone concentrations between
about 100 and 300 hPa indicated by sonde and MLS measurements. Furthermore, the retrieved profiles overestimate the
ozone concentration at the peak of the profile at about 100 hPa and underestimates the profile in the 7 to 28 hPa range
(Layers 5 and 6). The difference of –22.5 % between the retrieval and MLS 1 seen in Fig. 2c for Layer 5 is one of the largest
negative biases of all profiles processed. This large bias may partially be caused by errors in the measurement vector due to
clouds (The photodiode used for cloud correction was not available on this day). The large deviation for Layer 0 of 52 % is
not surprising considering that this layer is only 1.8 km thick and the sensitivity of the Umkehr method to ozone
concentrations close to the surface is poor.
The bias of the retrieval becomes smaller when the forward model is corrected for the systematic error resulting from the
pseudospherical approximation (compare Fig. 2c and Fig. 2g), indicating that the correction is appropriate.
The smallest difference between the retrieval on one hand and sonde and MLS measurements on the other is observed for
$\sigma_a$ = 0.1 (Fig. 2k). This suggests that a relatively small value for $\sigma_a$ is advantageous even though the sample profile
deviates considerably from the *a priori* profile. For Layers 5 to 9, the magnitude of the bias is comparable in magnitude to
the difference between the two MLS profiles, suggesting that a portion of the bias could be due to changes in the ozone
profile occurring during the period of Umkehr observations.
When $\sigma_a$ is set to 0.4, the RAKs of Layers 3 to 7 peak at the correct layer and drop to zero within two layers, suggesting
that ozone concentrations in this altitude range can be well resolved (Fig. 2d, h). In contrast, RAKs for layers 0, 1, and 2 are



similar and peak at about the same altitude. Hence, ozone concentrations in these layers cannot be separated well. The
altitude resolution of the standard Umkehr method is also poor in these layers, and results for layers 0 and 1 are typically
combined when reporting data. RAKs for layers $8 - 10$ peak at the same altitude, indicating that ozone concentrations above
the 3 hPa level (about 45 km) cannot be resolved and the retrieval is predominantly driven by the *a priori* profile. This is not
surprising considering the small ozone concentrations in these layers. Also the traditional Umkehr method has little
sensitivity at these altitudes.
When $\sigma_a$ is set to 0.1, the RAKs become rather broad (Fig. 2l). The solution is therefore more determined by the *a priori*
profile than the observations. The reduced importance of the measurements is also reflected in the value of $d_s$ : $d_s$ is 3.02
for $\sigma_a = 0.4$ and 2.15 for $\sigma_a = 0.1$.
TOCs calculated form the retrieved profiles agree well with the OMI measurements and depend only little on the choice of
retrieval parameters: absolute and relative biases are 7 DU (1.5 %) for parameter set (1), 6 DU (1.3 %) for set (2), and 1 DU
(0.2 %) for set (3).

## 3.1.2 Comparison for 11 April 2007

Fig. 3 shows results for 11 April 2007. On this day, ozone concentrations measured by sonde and MLS were consistently
below the *a priori* profile between 5 and 100 hPa, but between 100 and 300 hPa, the actual profile exceeded the *a priori*. Fig.
$3a - c$ show results calculated with $\sigma_a = 0.4$ while calculations used for Fig. $3c - f$ used $\sigma_a = 0.1$. The forward model was
corrected by scaling with $1/R(\theta)$ in both cases. Note that the MLS 1 and MLS 2 datasets are almost identical, indicating that
the actual ozone profile was constant over the observation period. RAKs are very similar to those for 19 April 2014 (Fig. 2h
and l) and are therefore not shown.
For both settings of $\sigma_a$, the retrieved profile is narrower than the *a priori* profile and matches the MLS profile almost
ideally for Layers $3 - 9$. This is an example that the retrieval result is not simply the *a priori* profile scaled with a constant
factor. Instead, the information contained in the measurement vector is sufficient to modify the shape of the profile to match
the actual, narrower shape. However, like in the case of the first example, the resolution of the Umkehr method is not
sufficient to reproduce the fluctuation of the actual ozone profile between 70 and 300 hPa. The most obvious difference
between the results calculated with $\sigma_a = 0.4$ and 0.1 is the difference at 183 hPa (Layer 2). Because the Umkehr method has
little sensitivity at this pressure level, the retrieved ozone concentration is mostly determined by the *a priori* profile for $\sigma_a =$
0.1 (Fig. 3f). In contrast, when setting $\sigma_a = 0.4$, measurements "pull" the retrieval to the higher concentrations of the actual
profile, resulting in a smaller bias relative to sonde and MLS data (Fig. 3c). The TOCs of both retrievals agree to within 7
DU (or 2.1 %) with OMI.





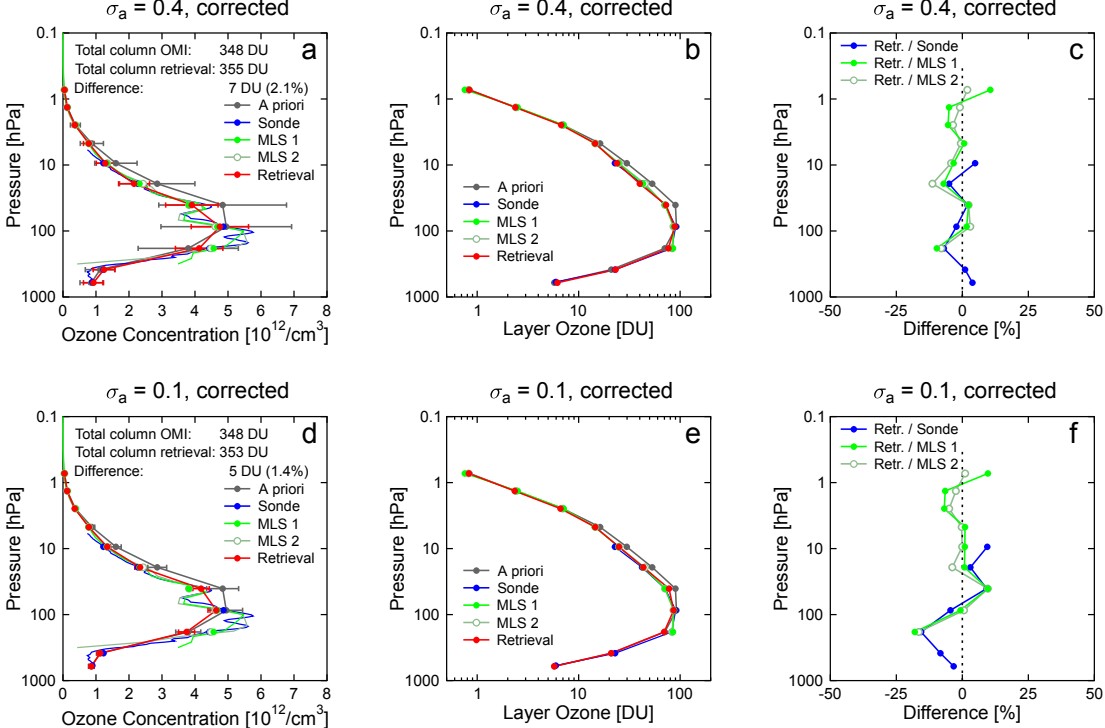

Fig. 3. Validation of ozone profile retrieved for 11 April 2007. Top row: results of $\sigma_a$ = 0.4 and corrected forward model.

Bottom row: $\sigma_a$ = 0.1, and corrected forward model. 1st column: ozone concentration as a function of pressure. 2nd column:

layer ozone as a function of pressure. 3rd column: difference between the retrieval and sonde, MLS 1 and MLS 2 datasets

averaged over each Umkehr layer. Labelling of the different datasets is identical to that of Fig. 2.

### 3.1.3 Comparison for 14 August 2009

The third example (Fig. 4) shows results from 14 August 2009 when the ozone profile was almost identical with the fall *a priori* profile. Note that this *a priori* profile is considerably below that for spring (e.g., Fig. 3d). Calculations were performed with $\sigma_a$ = 0.1 and the corrected forward model. Results agree with sonde and MLS profiles to within ±13 % for Layers 1 – 10 and the TOC of the retrieval is identical to the OMI measurement.

In summary, Umkehr profiles replicate the general pattern in the sonde and MLS data but cannot resolve the fine structure in the ozone distribution, in particular below 100 hPA. The relatively poor resolution in the troposphere and lower stratosphere is similar for the standard Umkehr method.





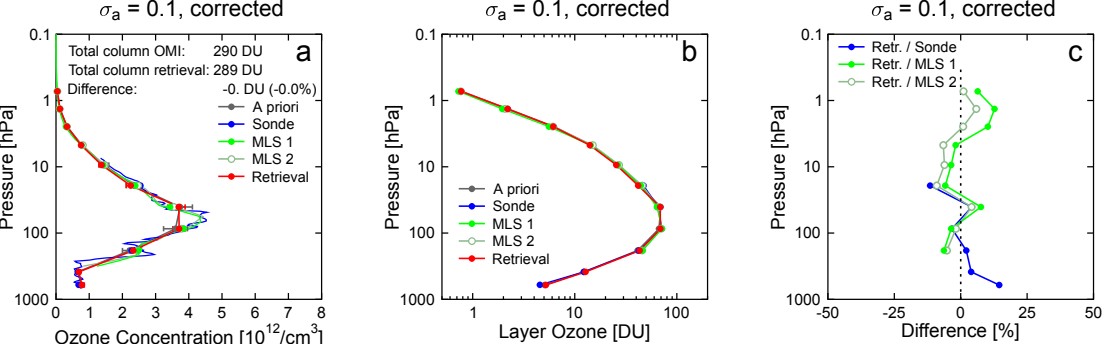

Fig. 4. Validation of ozone profile retrieved for 14 August 2009. The retrieved profile was calculated with $\sigma_a = 0.1$ using

the corrected forward model. (a) ozone concentration. (b) layer ozone. (c) difference between the retrieval and sonde, MLS 1

and MLS 2 datasets averaged over each Umkehr layer. Labelling of the different datasets is identical to that of Fig. 2.

**3.2 Comparison with balloon sonde and MLS profiles – statistics**

While the results for the three profiles discussed above are promising, they do not allow to assess the Global-Umkehr

technique comprehensively. To fully validate the method, we compared a large number of sonde and MLS profiles with our

retrievals using measurements from the years 2004 to 2014, and calculated statistics. We only considered periods when the

TOC was constant to within ±20 DU as indicated by OMI. This criterion restricted the number of comparisons with sonde

profiles to 57 and with MLS profiles to 552. Data were processed with and without the model correction discussed in Sect.

2.3 and with and without the cloud correction discussed in Sect. 2.2. The latter correction requires measurements of the

photodiode internal to the SUV-150B instrument. Unfortunately, these measurements were not available during all days,

reducing the number of retrieval/sonde and retrieval/MLS comparisons to 38 and 396, respectively. Results from Layers 0

and 1 and Layers 2 and 3 were combined because of the poor vertical resolution of the Umkehr methods in the troposphere

and lower stratosphere discussed earlier. Differences between retrieval and sonde, MLS 1, and MLS 2 data are illustrated

with box-whisker plots (Fig. 5), which show the minimum and maximum difference (black dots), median (black line),

average (red dot) interquartile (i.e., $25^{th} - 75^{th}$ percentile) range (box) and the $10^{th} - 90^{th}$ percentile range (whiskers) for each

layer or combination of layers. We also plotted statistics for the difference of the MLS 1 and MLS 2 datasets to indicate the

variability of the actual ozone profile over the course of one day. Fig. 5 includes results from spring and fall combined.

Table 2 provides statistics calculated separately for spring and fall.










Fig. 5. Box-whisker plots showing the difference between Umkehr retrieval results and sonde measurements (1st column),
MLS observations for day of retrieval ('MLS 1' dataset, 2nd column), and MLS observations for the following day ('MLS 2'
dataset, 3rd column). The 4th column illustrates the difference of the 'MLS 2' and 'MLS 1' datasets. Each plot shows the
minimum and maximum difference (black dots), median (black line), average (red dot) interquartile range (box) and the 10th
– 90th percentile range (whiskers) for each layer. Results for Layers 0 and 1 and Layers 2 and 3 were combined. The N-
number in the headers of each plot indicates the number of profiles used for computing the statistics. Results in each row
were calculated with a different set of parameters: 1st row (panels a–d): forward model not corrected, no cloud correction,
$\sigma_a = 0.4$. 2nd row (panels e–h): forward model corrected by scaling with $1/R(\theta)$, no cloud correction, $\sigma_a = 0.4$. 3rd row
(panels i–l): forward model corrected, no cloud correction, $\sigma_a = 0.1$. 4th row (panels m–p): forward model corrected, cloud
correction using data of photodiode, $\sigma_a = 0.1$.
Table 2. Bias and interquartile range (in parenthesis) of retrieval/MLS 1 comparison; average and standard deviation of the
difference between total ozone calculated from retrieved profiles and measured by OMI (TOC); and average number of
degrees of freedom for signal ($<d_S>$) for spring and fall periods. The second column provides the number of profiles $N$
contributing to the statistics.

| Season | $N$ | Bias and interquartile range of retrieval/MLS 1 comparison for Layer | | | | | | | | TOC | $<d_s>$ |
|---|---|---|---|---|---|---|---|---|---|---|---|
| | | 2 & 3 | 4 | 5 | 6 | 7 | 8 | 9 | 10 | | |
| *No model correction, no cloud correction, $\sigma_a = 0.4$* | | | | | | | | | | | |
| Spring | 197 | 4% (14%) | -1% (10%) | -8% (9%) | -10% (8%) | -4% (9%) | -1% (11%) | 0% (10%) | 4% (15%) | 0.2% (1.9%) | 3.1 |
| Fall | 355 | 6% (10%) | -1% (12%) | -8% (12%) | -7% (9%) | -3% (7%) | 1% (9%) | 1% (11%) | 3% (11%) | 0.7% (1.8%) | 3.0 |
| *Model correction, no cloud correction, $\sigma_a = 0.4$* | | | | | | | | | | | |
| Spring | 197 | 2% (13%) | -1% (10%) | -6% (10%) | -6% (8%) | -4% (10%) | -5% (11%) | -5% (10%) | 0% (14%) | 0.0% (1.9%) | 3.1 |
| Fall | 355 | 4% (10%) | -1% (12%) | -4% (13%) | -4% (9%) | -4% (7%) | -3% (9%) | -3% (11%) | 0% (11%) | 0.5% (1.8%) | 3.0 |
| *Model correction, cloud correction, $\sigma_a = 0.4$* | | | | | | | | | | | |
| Spring | 142 | 3% (13%) | -2% (12%) | -6% (10%) | -6% (8%) | -5% (10%) | -6% (12%) | -6% (10%) | -1% (15%) | 0.0% (2.0%) | 3.1 |
| Fall | 254 | 3% (10%) | -1% (11%) | -4% (12%) | -3% (9%) | -4% (8%) | -3% (10%) | -3% (10%) | -1% (11%) | 0.5% (1.9%) | 3.0 |
| *Model correction, no cloud correction, $\sigma_a = 0.1$* | | | | | | | | | | | |
| Spring | 197 | 1% (12%) | -1% (12%) | -4% (12%) | -5% (10%) | -4% (10%) | -4% (12%) | -4% (13%) | 1% (14%) | -0.2% (1.8%) | 2.2 |
| Fall | 355 | 2% (9%) | 0% (12%) | -3% (14%) | -3% (8%) | -4% (7%) | -3% (9%) | -3% (12%) | 0% (12%) | 0.3% (1.7%) | 2.1 |
| *Model correction, cloud correction, $\sigma_a = 0.1$* | | | | | | | | | | | |
| Spring | 142 | 1% (12%) | -1% (13%) | -4% (12%) | -5% (10%) | -4% (11%) | -4% (12%) | -5% (13%) | -1% (16%) | -0.2% (1.9%) | 2.2 |
| Fall | 254 | 2% (8%) | -1% (11%) | -2% (14%) | -2% (8%) | -3% (7%) | -4% (9%) | -4% (12%) | -1% (11%) | 0.3% (1.7%) | 2.1 |



The 1$^{st}$ row (panels a–d) of Fig. 5 shows results calculated without the model and cloud corrections; $\sigma_a$ was set to 0.4. The

average and median biases between retrieval and MLS data vary between –8 % and +5 % (Fig. 5b, c). The largest negative

bias is observed for Layers 5 and 6 while the largest positive bias of 5 % is observed closest to the surface (Layer 2&3).

Biases relative to the sonde measurements (Fig. 5a) are by and large consistent with biases relative to MLS data, although

the comparatively small number of sonde observations makes statistics less robust. Fig. 5d confirms that there is no

systematic difference between the MLS measurements on the day of Umkehr observations (MLS 1) and the following day

(MLS 2).

For the retrieval/MLS comparisons, the interquartile ranges vary between 7 % and 12 % and depends only modestly on the

layer. With the exception of the results for the highest layer, the interquartile ranges for the MLS 2 to MLS 1 comparison

vary between 5 % and 10 %. Differences between the 10$^{th}$ and 90$^{th}$ percentiles vary between 14 % and 24 % for the

retrieval/MLS comparisons (whiskers in Fig. 5b, c) and between 12 % and 17 % for the MLS 2 / MLS 1 comparison,

excluding the highest layer (Fig. 5d). The similarity of the ranges for the retrieval/MLS and MLS 2 / MLS 1 comparisons

suggests that a large portion of the observed retrieval/MLS differences can be attributed to changes in the actual ozone

profile over the time periods relevant for these comparisons. Lastly, the large interquartile range for the retrieval/sonde

comparison observed in Layer 0&1 (Fig. 5a) is again a manifestation of the fact that the Umkehr method has little sensitivity

for the layers closest to the surface.

To assess the effect of the forward model correction on our Umkehr retrievals, we repeated the calculations with this

correction. Results are presented in the 2$^{nd}$ row (panels e–h) of Fig. 5. As before, no cloud correction was applied and $\sigma_a$

was set to 0.4. By comparing the original results (Fig. 5b, c) with the corrected results (Fig. 5f, g) it can be observed that the

bias between retrieval and MLS data has diminished and now varies between –5 % (Layers 5 and 6) and +3 % (Layer 2&3),

suggesting that the model correction is justified. The interquartile ranges with and without the correction are virtually

indistinguishable. Note that the correction has no effect on the MLS 2 / MLS 1 comparison and Fig. 5d and h are therefore

identical.

To explore the effect of $\sigma_a$ on the results, we repeated the calculations using $\sigma_a$ = 0.1 instead of $\sigma_a$ = 0.4. Results are

shown in the 3$^{rd}$ row (panels i–l) of Fig. 5. For $\sigma_a$ = 0.1, the bias between retrieval and MLS data decreased further and now

varies between –4 % and +1 % (Fig. 5j, k). Differences between retrieval and sondes (Fig. 5i) have also decreased compared

to calculations with $\sigma_a$ = 0.4, except for Layer 0&1. The observation that biases are larger for a larger value of $\sigma_a$ could be

caused by systematic errors in the measurement vector or an incomplete correction of the forward model results. Changing

$\sigma_a$ from 0.4 to 0.1 had almost no effect on the interquartile range, however, minimum and maximum differences (black

dots) contracted somewhat.





Finally, the calculations were repeated with the cloud correction turned on (4th row of Fig. 5, panels m–p). For the retrieval/MLS comparison, biases and interquartile ranges with and without the cloud correction agree to within 1 %. Results for the retrieval to sonde comparison (Fig. 5m) are affected by the small sample size of N=38. (Note that results shown for Layer 6 are only based on eight samples because most balloons burst before they reach this layer).

The difference between uncorrected and cloud-corrected statistics is very small, suggesting that clouds affect the accuracy of the retrievals only marginally. However, this conclusion may not apply to locations with thicker clouds and should be tested if the method is used at other sites.

Table 2 allows to assess retrievals for spring and fall periods separately. Because statistics are more robust for the retrieval/MLS than retrieval/sonde comparisons, Table 2 only presents results for the former. Biases and interquartile ranges

are provided with and without the model and cloud corrections, and with $\sigma_a$ set to either 0.4 or 0.1. Biases for spring and fall agree to within 3 % for all layers. When no corrections are applied and $\sigma_a$ = 0.4, biases range between –10 % (Layer 6 for spring) to +6 % (Layer 2&3 for fall). The model correction decreases this range to –6 % to 4 %. By reducing $\sigma_a$ from 0.4 to 0.1, the range decreases further to –5 % to 2 %. The cloud correction has a negligible (≤ 1 %) effect on the biases. Interquartile ranges vary from 7 % to 16 % and depend only little (≤ 3 %) on $\sigma_a$ and on whether or not corrections are

applied.

Table 2 also includes a column comparing TOCs derived from the retrieved profiles with measurements by OMI. Depending on $\sigma_a$ and the correction method, the average difference between the retrieved and OMI TOCs varies between –0.2 % and 0.7 %, and the standard deviation is varies between 1.7 % and 2.0 %.

Lastly, the average value of $d_s$ is about 3.0 for $\sigma_a$ = 0.4 and 2.1 for $\sigma_a$ = 0.1. A value of $d_s$=3.0 may seem low, but is

consistent with values of $d_s$ resulting from the standard, zenith-sky radiance Umkehr technique. For example, Stone et al. (2015) reported a value of $d_s$=3.1 for Dobson Umkehr retrievals using the Dobson C wavelength pair (311.4 and 332.4 nm) and the standard Dobson SZAs ranging from 60° to 90°.

## 4 Discussion

When the forward model is corrected, the bias of our retrievals relative to MLS data is smaller than ±6 % for all layers. This

level of agreement compares favourably with published results of the standard zenith sky Umkehr method. For example, McElroy and Kerr (1995) compare Umkehr profiles derived from a Brewer spectrophotometer with concurrent measurements of a lidar, a microwave radiometer and ozone sondes, which were performed during a one-month campaign at the Table Mountain Observatory in California. The mean bias between the Brewer Umkehr results and the mean of the other instruments varied to within ±10 % for altitudes between 20 and 35 km. Between 37 and 47 km, the Brewer data were low

by 15 to 20 % (Fig. 9 of McElroy and Kerr, 1995).



Nair et al. (2011) compare stratospheric ozone vertical distribution measured by a large number of ground-based and satellite sensors at the Haute-Provence Observatory, France. They find that zenith-sky Umkehr data from an automated Dobson spectrophotometer systematically underestimate the stratospheric ozone concentration with a near-zero bias at about 30 km, but increasing to 7 % at 21 km and 34 km, and to 14 % at 40 km (Fig. 8 of Nair et al., 2011). Despite of these large biases,

Nair et al. (2011) conclude that Umkehr data are useful for studies of the long-term ozone evolution and for detecting drifts in satellite observations.

Miyagawa et al. (2014) compare Dobson Umkehr measurements with homogenized NOAA SBUV (Solar Backscatter Ultraviolet Instrument)(/2) 8.6 overpass data measured between 1977 and 2011. The mean bias between Dobson and SBUV partial ozone column varied between -12 % for Layer 7 to +3 % for Layer 2 (Fig. 1a of Miyagawa et al., 2014).

The biases reported in the three studies quoted above are comparable or larger than the differences between our Umkehr retrievals and MLS and sonde measurements, suggesting that Umkehr results derived from global spectral irradiance can provide data of similar accuracy than the established zenith-sky method. A portion of the retrieval/MLS difference could also be caused by systematic errors in the MLS dataset, considering that the MLS accuracy specified by Froidevaux et al. (2008) is in the 5 to 10 % range.

Results presented in Fig. 5 illustrate that interquartile and $10^{th} – 90^{th}$ percentile ranges for the retrieval/MLS comparison on one hand and the MLS 2 / MLS 1 comparison on the other are similar for most layers. This suggests that a large portion of the observed retrieval/MLS differences can be attributed to actual changes in the ozone profile over the time periods relevant for these comparisons. However, a portion of the change in the MLS profile from one day to the next may be caused by the relatively poor horizontal resolution of MLS profiles of about 200 km. For example, some variability in the MLS overpass

dataset can be attributed to the slightly different geolocation of two consecutive overpass profiles.

Another source of variability in the retrieval/MLS and retrieval/sonde comparisons is the different vertical resolutions of MLS (about 2-3 km), sondes (0.1 km) and our Umkehr retrievals (about 10 km for $\sigma_a = 0.4$ and about 25 km for $\sigma_a = 0.1$). If measurement and forward model were without error, an Umkehr profile would resembles the actual profile smoothed by the AKs. To reduce the effect of the differing resolution, the higher-resolution MLS profiles could be convolved with the

AKs of the Umkehr profile prior to comparing the two profiles. This technique has for example been applied by Nair et al. (2011) when comparing lidar and SBUV profiles. We did not use this method because it artificially reduces the true difference that is observed when comparing a high-resolution (sonde, MLS) with a low-resolution (Umkehr) profile. Of note, Nair et al. (2011) found that the smoothing technique does not make a significant difference in seasonally averaged data such as those presented in Fig. 5 and Table 2.

The bias between Umkehr retrievals and MLS or sonde data is reduced when correcting the forward model for the systematic error presented by the pseudospherical approximation. It is interesting to note that the correction is only in the –0.5 to 2.0 % range (Fig. 1b) but reduces the retrieval bias by up to 4 % (Layer 6 in spring, see Table 2). Considering that the uncertainty of our measurements is 3 % (1σ), systematic errors in the measurement vector in the 2-3 % range could conceivably be responsible for the remaining bias of Umkehr and MLS profiles indicated in Fig. 5 and Table 2. To test this hypothesis, we



modified the measurement vector within reasonable limits and recalculated the profiles. We found that the bias between of Umkehr and MLS profiles cannot be significantly reduced further, suggesting that the bias cannot be attributed to measurement errors alone.

The difference between results corrected for cloud effects and uncorrected results is very small, implying that clouds affect
the accuracy of the retrievals only marginally. However, this conclusion may not apply to locations with thicker clouds or locations affected by aerosols and should be tested if the method is used at other sites.

If $\mathbf{S}_\varepsilon$ is well defined, the most important parameter to optimize the results is $\sigma_a$. The objective is to find the right balance between sensitivity to the *a priori* profile on one hand and sensitivity to (unavoidable) errors in the measurement vector or forward model on the other. We chose $\sigma_a$ = 0.1 and 0.4. The smaller value quantifies the standard deviation of the actual
variability of the ozone profile at Summit. While calculations with this value lead to good results, the solution may not be optimal for profiles at the fringe of the distribution (e.g., result for Layer 3 in Fig. 3). A small $\sigma_a$ also results in a small value of $d_s$. However, statistics for results calculated with $\sigma_a$ = 0.1 and 0.4 are quite similar (Table 2), suggesting that any value for $\sigma_a$ between 0.1 and 0.4 leads to acceptable profiles. Determining the best value for sites other than Summit requires consideration of the measurement system and variability of the ozone profile at this site.

There are various ways to optimize the Global-Umkehr method for specific applications or locations. For example, if two instruments were to measure side by side, the uncertainty used to set up $\mathbf{S}_\varepsilon$ could be better estimated by comparing the measurements of the two systems. Furthermore, the method to set up $\mathbf{S}_a$ could be modified to take into account that the variability of the ozone profile depends on altitude (Eq. (4) uses the same standard deviation $\sigma_a$ for all layers). Finally, SZAs and wavelengths used for the measurement vector could be further optimized to reduce uncertainties related to the
Ring effect or the temperature dependence of the ozone absorption cross section.

We used *a priori* profiles that are independent of the total ozone column. Umkehr retrievals from Dobson instruments that have historically been processed with the algorithm developed by Mateer and DeLuisi (1992) used TOC-dependent *a priori* profiles to constrain the retrieval. While this practice can lead to artefacts when calculating trends (Petropavlovskikh et al., 2005; Stone et al., 2015), the approach may be the best choice if a profile with the smallest uncertainty possible is sought for
a specific purpose.

The Global-Umkehr method was tested with spectroradiometric measurements from a polar location because we only operate instruments at high-latitude sites. Inversions using high-latitude data are more challenging compared to retrievals for lower latitudes because of the limited range of SZAs at polar regions, the long time that is required to scan the range of SZAs necessary for the retrieval, and the high short- and long-term variability of the ozone profile. We have therefore confidence
that the method would work well for mid- and low-latitude locations. Confirmation of this assertion is subject of future tests.





## 5 Conclusions

An optimal estimation method has been developed to retrieve vertical ozone profiles from measurements of global spectral irradiance in the UV. The method is similar to the widely used Umkehr technique, which inverts measurements of zenith sky radiance. To our knowledge, this is the first time that the Umkehr technique was applied to measurements of global

irradiance. High-quality measurements of global spectral irradiance are now available for more than 25 years at several NDACC locations (De Mazière et al., 2017), and the Global-Umkehr method has the potential to make these long-term datasets available for studying changes in the vertical distribution of ozone.

Compared to the standard zenith sky Umkehr method, multiple scattering effects have to taken into consideration when exploiting global irradiance measurements, which also include contributions from photons received from directions close to

the horizon. We have shown that this challenge can be overcome by using a forward model with pseudospherical approximation plus additional corrections.

The method was evaluated with spectroradiometric measurements from Summit, Greenland, and validated with balloon sonde and MLS observations. For calculations using the corrected forward model, the bias between our retrieved profiles and MLS observations ranges between −5 % (Layers 5 and 6) and +3 % (Layer 2&3). The magnitude of this bias is comparable,

if not smaller, to values reported in the literature for the standard Umkehr method. The distribution of the difference between retrieval and MLS observations was quantified with the interquartile and $10^{th} – 90^{th}$ percentile ranges. Depending on altitude, the interquartile ranges vary between 7 % and 13 % and the $10^{th} – 90^{th}$ percentile ranges run between 14 % and 24 %. Of note, interquartile ranges calculated from the differences of two MLS profiles that were measured on consecutive days vary between 5 % and 10 %, suggesting that a considerable portion of the retrieval/MLS differences can be attributed to real

changes in the ozone profile. For Umkehr Layer 2 and higher, retrieval/MLS and retrieval/sonde differences are by and large consistent. The poor sensitivity of the Umkehr method to the altitude range of Layer 0&1 leads to are relatively large scatter (e.g., the interquartile range is 25 %) of the retrieval/sonde differences for this layer.

The effect of the parameter $\sigma_a$, which controls the sensitivity of the solution on the *a priori* profile, was extensively assessed. It was found that results calculated with a small value of $\sigma_a = 0.1$ (emphasis on *a priori*) generally agree to within

2-3 % with those calculated with a large value of $\sigma_a = 0.4$ (emphasis on measurements). By setting $\sigma_a$ to a large value, retrieval errors may occasionally become large if the measurement vector is affected by unforeseen conditions (e.g., changing ozone, variable clouds). For example, the maximum retrieval/MLS difference was 50 % for $\sigma_a = 0.4$ but only 32 % for $\sigma_a = 0.1$.

The retrieved ozone profiles were integrated over altitude. The resulting TOCs agreed almost ideally with TOCs measured

by OMI: depending on the correction method, the retrieval/OMI bias ranged between -0.2 % and 0.7 % with a standard deviation of less than 2.0 %.



While the Global-Umkehr method was only tested for a high-latitude site, we are confident that it will also work at lower latitudes, but this assertion requires confirmation by future tests.

**Data availability**

"Version 2" spectra from the SUV-150B spectroradiometer at Summit are available from the Arctic Data Center at
https://arcticdata.io/.

*Acknowledgements.* Funding for this study was provided by the US National Science Foundation's Office of Polar Programs Arctic Sciences Section (award ARC-1203250, Ultraviolet Radiation in the Arctic: 2012-2015). We are grateful to the numerous dedicated individuals who have operated the UV radiometers at Summit. We also thank Bryan Johnson of NOAA
for providing ozone profiles collected by balloon ozonesondes that were launched at Summit on weekly bases. NSF provided funding for the ozonesonde measurements at Summit.

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
