# Peer review of "Retrieving Vertical Ozone Profiles from Measurements of Global Spectral Irradiance"

_Atmospheric Measurement Techniques, 2017_

## Referee Comment (RC1) · Anonymous Referee #1 · 18 Aug 2017

General comments

This paper describes a new technique to retrieve vertical resolved ozone from direct sun and upper hemisphere ultraviolet (zenith to pole) measurements. It includes validation of the retrievals with the Microwave Limb Sounder, Ozone Monitoring Instrument, and ozonesonde measurements as well as discussion of comparison of retrieval biases with similar methods such as the standard zenith sky Umkehr technique.

This is a very good paper that is well within the scope of AMT, well written, and will be of general interest to the ozone community, especially those that have similar data available. My main comments are: considering that this paper may lead to retrieval of more historical ozone information in addition to what is already available (ozonesondes, Dobson Umkehr, etc.). I would like to see a bit more discussion on the temporal and

spatial availability of the measurements. Consistent (Zenith-sky and Global-Umkehr) naming conventions would be nice to avoid any confusion. I would also like to see some discussion of the local times that MLS passes over summit and seasonal change in the averaging kernel (if any). I recommend publication of this paper after those and the following clarifications and changes.

Specific comments

Page 1, line 26: I would like to see a short explanation of what this measurement (the direct sun plus upper hemisphere) is typically used for.

Page 2, line 2 and line 21 and Section 2.2: You mention that there are other sites that have these UV detectors. It would be nice to have some general information about how many potential sites there are, their temporal measurement range, are there any Southern Hemisphere sites, and there many locations where there are not any Umkehr measurements, etc. It is briefly mentioned in the conclusions that there are many sites with time series of greater than 25 years, but this is not mentioned anywhere else.

Page 3, line 4: Why not use the more recent ozone cross section studies? https://www.atmos-meas-tech.net/7/609/2014/

Page 4, line 19: You mention on line 14 that a $\sigma$_a value of .1 is small and therefore very sensitive to the a priori. However, you go on to say on line 19 that $\sim$.1 is the standard deviation of the MLS profiles. I feel this needs clarification as you mention that $\sigma$_a is the anticipated variability (standard deviation) and therefore using a value higher than .1 (for example .4) means you are expecting a larger variability in the retrieval.

Page 10, line 1: What two times of day are MLS measurements taken at the latitude of Summit? Are there any inconsistencies here, diurnal effects, polarisation?, etc?

Page 15, Fig. 4: It would be interesting to see if the change in season (thus, the vertical structure of the ozone profile) modifies the structure of the relative averaging kernels, especially, as fall and spring statistics are compared later on in Table 2.

Page 15, Fig 4: Also, why are the $\sigma\_a$ = .4 plots not shown in this figure? It would be interesting to see if the inversion agrees well in this case when it has more freedom due to a larger a priori covariance. If you have the results, they could also just be mentioned in the text.

At the beginning of the paper, you define Umkehr to refer to the standard zenith sky Umkehr technique and Global-Umkehr to refer to direct sun plus upper hemisphere. However, throughout the text and especially in the discussion you refer to Global-Umkehr as just Umkehr which is confusing. I suggest keeping the naming conventions consistent throughout the text.

Technical corrections

Page 1, line 11: Substitute ultraviolet for UV.

Page 1, line 18: The OMI acronym does not need to be included here as it is not repeated in the abstract. It is redefined in the main text.

Page 2, line 4: Double closed bracket.

Page 7, line 2: Is the AFGL acronym defined (Air Force Geophysics Laboratory)?

Page 14, line 11: suggest changing identical to virtually identical as there is a small difference of 1 DU as seen in Figure 4.

Page 15, line 7: Confusing sentence, suggest to change: ...they do not allow to assess the Global-Umkehr technique comprehensively. to something like they do not allow the comprehensive assessment of the Global-Umkehr technique.

Page 5, line 10: Spaces seem to be present between all equations and symbols and full stops, commas. This can be misleading in some instances. For example, Page 5, line 10 may be interpreted as a dot product.

Table 1. There are spaces on either side of the endashes which are not consistent with endash ranges throughout the text.

Page 10, line 1: typo - MLS measure(s) thermal... - remove "s"

Page 10, line 15: space after second open bracket.

Page 10, line 24: Suggest remove therefore or move to the start of the sentence - Therefore,...

Page 17, line 15: should N be in parenthesis?

Page 19, line 8: Change Table 2 allows to assess retrievals... to something like Table 2 allows the assessment of retrievals...

Page 19, lines 12 and 13: change to to between -6 % to 4 % and to between -5 % to 2 %

Page 19, line 18: remove is

Page 19, lines 19 and 21: insert a space after the equals sign

Page 19, line 19: Change to but it is consistent

Page 19, line 20: remove comma after standard

Page 20, line 8: Is (/2) meant to be there?

Page 20, line 23: change resembles to "resemble

Page 22, line 8: change to ...have to be...

Page 22, line 24: change to ...2–3 % of those... (use an endash?)

---

## Referee Comment (RC2) · Anonymous Referee #2 · 19 Sep 2017

General Comments:

This paper presented a new algorithm titled global-Umkehr algorithm to retrieve the vertical distribution of ozone from global spectral irradiances. The algorithm was tested using the data measured at Summit, Greenland and the retrievals were validated with ozonesonde and MLS data. The validation demonstrates good retrieval performance comparable to those from the standard Umkehr technique. As similar global irradiance measurements have been routinely measured at a number of stations, applying this technique to those measurements has the potential to complement the Umkehr measurements and contribute to the long-term monitoring of ozone profiles.

This paper is very suitable for publication in AMT. It is generally well written and organized except that some subsections of section 2 can be rearranged to further improve

the organization. The validations with ozonesonde and MLS data are very well presented and discussed. Overall, I recommend it to be published after addressing the following minor comments.

Specific Comments:

1. It is good to provide more details about how to derive a priori profiles from MLS and ozonesonde data. Are MLS data collocated with ozonesonde data around the Summit station? How MLS and ozonesonde data are merged as they cover different altitude ranges? Have other ozone profile climatologies such as McPeters et al. (2007) and McPeters and Labow (2012) been considered?

2. Instead of using fixed a priori error of 0.1 and 0.4, you mentioned the use of altitude-dependent a priori errors in the discussion (P21 L18), which is likely more appropriate as the ozone variability is relatively small in most of the stratosphere, $\sim$10% based on your analysis, but increases significantly in the lower stratosphere and upper troposphere to $\sim$40%. You can modify Eq 4 to be more generic, allowing for altitude-dependent a priori errors: $[Sa]_{mn} = sigma\_am^2 * [Xa]_m * sigma\_an^2 * [Xa]_n * exp(-|m-n|/d)$

3. P5, L8, one of the most important diagnostics is averaging kernels A, which is described in section 2.4. I suggested moving section 2.4 to in front of L8 as $ds$, is typically derived from A, as the trace of A. The diagonal elements of A are the $ds$ at each layer.

4. P6, Equation 8 is confusing. Looks like $Dc(theta(t))$ is not based on actual measurement, but based on the parameterization of clearly sky measurement as a function of SZA. You may change "$Dc(theta(t))$ is the measurement ..." to "$Dc(theta(t))$ is the modeled photodiode measurement at time t that would be observed during clear skies, parameterized a function of SZA after filtering cloudy measurements." Also what criteria are used to filter cloudy measurements?

5. It is better to switch sections "Retrieval method" and "Measurements" as the section of retrieval method depends on the description of measurements.

6. P6, L18-21, what is the main motivation of interpolating measurements to a common SZA grid that has 8 SZAs other than reducing the computation time. What is the typical number of spectra during the collection period (SZA change form 70 to 90)? Looks like it is much larger than 8, so interpolating it to 8 SZAs only while keeping the same measurement error can reduce the available information content and increase the measurement error. Have retrievals been conducted using the measurements at individual SZAs and compared with retrievals interpolated to 8 common SZAs?

7. P6, L30, why not using more recent ozone cross sections based on the activities of ACSO (Absorption Cross-Sections of Ozone) summarized in Orphal et al. (2016), which recommends that the BP data should not be used. Is this for consistency with the OMI TOC retrieval, which also used the BP data?

8. Are both SDISORT and MYSTIC RTMs based on scalar (rather than vector) radiative transfer models? If so, this is another source of forward model bias. What are the impacts of neglecting polarization (i.e., assuming scalar) on the calculated radiances? Just check if any such analysis has been done for either SIDOSRT and MYSTIC RTM.

9. P10, L12, how is this threshold of 20 DU be determined?

10. P11, MLS measurements from consecutive days are used to quantify the temporal variation of ozone. It should be noted that the MLS measurements from consecutive days will be measured at different locations, maybe ∼100 km apart. So some of the MLS1/2 difference is due to spatial variability. What is the average distance between MLS 1 and 2?

11. P12, L3-15, a lot of the description can be reduced as this has been described in the figure caption.

12. P21 L20, you may consider using some recent cross sections as suggested in Orphal et al. (2016), and use meteorological data (e.g., temperature profiles) to account for the temperature dependence of the ozone absorption cross section. To reduce the impact of Ring effect, you may consider optimizing not only wavelengths, but also the magnitude of bandpass (currently 2 nm) used to degrade the spectral resolution. In addition, you can also mention the correction of forward model errors due to the neglect of polarization as commented earlier.

13. P22 L8, multiple scattering effect is also important for zenith sky measurements. You may say multiple scatterings effects become more important and the sphericity of the viewing geometry should be taken into account.

14. P24, L21, The poor sensitivity of the Umkehr method to ozone retrieval at layer 0 & 1 was mentioned here. Because only 2 wavelengths are used in the retrievals, measurements at other wavelengths especially the global irradiance spectrum can be used to improve the retrieval sensitivity in the first few layers as shown in Liu et al. (2005). You may add a few sentences about the possibility of exploring this for future studies.

Technical comments

1. P2, L20, change "a.s.l" to "a.s.l."

2. The section of "1 Method" should be "2 Method" and "1.1 Retrieval method" should be "2.1 Retrieval method"

3. P2, L25, change "depends" to "depend"

4. P5, L17, "and is part of..."

5. P5, L27, change "wavelengths shifts" to "wavelength shifts"

6. P6, L17, this sentence can be grouped to the above paragraph.

7. P6, L28, change "result are" to "results are"

8. P7, L6, change "reference" to "references"

9. P7, L24, change "considered" to "considered as"

10. P15, L18, add "," before "interquartile"

11. P19, L28, change "decreased" to "has decreased" or "decreases"

12. P19, L18, change "is varies" to "varies"

13. P19 L26ïijŇ change to "compared"

14. P20, L1, L2, L5, L6, change to "compared", "found", "concluded", "compared"

15. P23 L10, change to "on a weekly basis"

16. P24, L17, add "," after "Phys."

17. P24, L18, add "," after "Res."

18. P26, last line, use normal font for the journal title.

McPeters, R. D., Labow, G. J., and Logan, J. A.: Ozone climatological profiles for satellite retrieval algorithms, J. Geophys. Res., 112, D05308, doi: 10.1029/2005jd006823, 2007. McPeters, R. D., and G. J. Labow (2012), Climatology 2011: An MLS and sonde derived ozone climatology for satellite retrieval algorithms, J. Geophys. Res., 117, D10303, doi:10.1029/2011JD017006.

Orphal, J., et al., Absorption cross-sections of Ozone in the ultraviolet and visible spectral regions – Status report 2015, J. Mol. Spectrosc., 327, 105-121, doi:10.1016/j.jms.2016.07.007, 2016.

Liu, X., K. Chance, C.E. Sioris, M.J. Newchurch, T.P. Kurosu, Tropospheric ozone profiles from a ground-based ultraviolet spectrometer: a new retrieval method, Appl. Opt., 45(10), 2352-2359, 2006.

---

## Author Comment (AC2) · 24 Oct 2017

**Response to comments of Referee #1**

We thank the referee for his or her thoughtful comments, which we have addressed as follows:

**Referee comments**
I would like to see a bit more discussion on the temporal and spatial availability of the measurements. Consistent (Zenith-sky and Global-Umkehr) naming conventions would be nice to avoid any confusion. I would also like to see some discussion of the local times that MLS passes over summit and seasonal change in the averaging kernel (if any).

**Response**
The new version of the manuscript now uses the terms "standard zenith-sky Umkehr method" and "Global-Umkehr method" consistently. Other issues noted in this general comment are discussed below.
* * *
**Referee comment**
Page 1, line 26: I would like to see a short explanation of what this measurement (the direct sun plus upper hemisphere) is typically used for.

**Response:**
The following was added to the manuscript:
"Such measurements were started by several groups in the early 1990s to monitor changes in UV radiation at the Earth's surface. These activities were motivated by concerns that decreases in atmospheric ozone concentrations, which were caused by ozone depleting substances released by man into the atmosphere, could lead to increases in UV radiation with detrimental effects on human health, and terrestrial and aquatic ecosystems (e.g., Bais et al., 2015)."

**The following reference was added:**
Bais, A. F., McKenzie, R. L., Bernhard, G., Aucamp, P. J., Ilyas, M., Madronich, S., and Tourpali K.: Ozone depletion and climate change: impacts on UV radiation, Photochem. Photobiol. Sci., 14(1), 19-52, 2015.
* * *
**Referee comments**
Considering that this paper may lead to retrieval of more historical ozone information in addition to what is already available (ozonesondes, Dobson Umkehr, etc.). I would like to see a bit more discussion on the temporal and spatial availability of the measurements.

and

Page 2, line 2 and line 21 and Section 2.2: You mention that there are other sites that have these UV detectors. It would be nice to have some general information about how many potential sites there are, their temporal measurement range, are there any Southern Hemisphere sites, and there many locations where there are not any Umkehr measurements, etc. It is briefly mentioned in the conclusions that there are many sites with time series of greater than 25 years, but this is not mentioned anywhere else.

**Response**
The majority of instruments that provide global spectral UV irradiance measurements suitable for the Global-Umkehr method are part of the UV monitoring networks mentioned in the introduction. While it is beyond the scope of this manuscript to assess the suitability of each of these systems, we estimate that about 25 instruments meet the accuracy requirements for the Global-Umkehr method and could potentially be utilized. The following was added to the manuscript:

"We estimate that about 25 spectroradiometers that are part of the various UV monitoring networks mentioned earlier provide data of sufficient quality for the Global-Umkehr method. Some of these instruments were established in the early 1990s at locations around the globe, including the Arctic, North America, Hawaii, Europe, New Zealand, Australia, and Antarctica."

We also added a link to the European UV Database, which also includes quality-controlled spectral UV measurements that are potentially suitable for the Global-Umkehr method.
* * *
**Referee comment**
Page 3, line 4: Why not use the more recent ozone cross section studies?
https://www.atmos-meas-tech.net/7/609/2014/

**Response**
We note that Referee 2 had a similar comment.
We are aware that more accurate ozone absorption cross sections than those published by Bass and Paur (1985) are now available and recommended. Nonetheless, we decided to use the Bass and Paur (1985) data because OMI total ozone data are based on B&P. Using a different cross section would have complicated the validation of our results with OMI data. We added the following to the manuscript:
"While more accurate ozone absorption cross sections are now available (Gorshelev et al., 2014; Orphal et al., 2016), we used Bass and Paur (1985) data to facilitate validation with OMI total ozone column measurements, which are also based on Bass and Paur (1985)."

**The following references were added:**

Gorshelev V., Serdyuchenko A., Weber M., Chehade W., and Burrows J.P.: High spectral resolution ozone absorption cross-sections–Part 1: Measurements, data analysis and comparison with previous measurements around 293K, Atmos. Meas. Tech., 7, 609-624, doi:10.5194/amt-7-609-2014, 2014.
Orphal J., Staehelin J., Tamminen J., Braathen G., De Backer M. R., Bais A., Balis D., Barbe A., Bhartia P. K., Birk M., and Burkholder J. B.: Absorption cross-sections of ozone in the ultraviolet and visible spectral regions: Status report 2015, J. Mol. Spectrosc., 327, 105-121, doi:10.1016/j.jms.2016.07.007, 2016.
* * *
**Referee comment**
Page 4, line 19: You mention on line 14 that a $\sigma_a$ value of .1 is small and therefore very sensitive to the a priori. However, you go on to say on line 19 that ~.1 is the standard deviation of the MLS profiles. I feel this needs clarification as you mention that $\sigma_a$ is the anticipated variability (standard deviation) and therefore using a value higher than .1 (for example .4) means you are expecting a larger variability in the retrieval.

**Response**
As discussed in great detail in the manuscript, $\sigma_a$ is an important parameter to optimize the solution. At the onset of the study, the optimal value for $\sigma_a$ was not known, although a value of 0.1 is supported by the standard deviation of the MLS profiles. We therefore performed calculations with two settings, 0.1 and 0.4. We feel that the pros and cons of using either 0.1 or 0.4 are discussed in sufficient detail (in particular in the "Discussion" section), and think that lengthening the discussion further is not necessary. No change to manuscript.
* * *
**Referee comments**
I would also like to see some discussion of the local times that MLS passes over summit.

and

Page 10, line 1: What two times of day are MLS measurements taken at the latitude of Summit? Are there any inconsistencies here, diurnal effects, polarisation?, etc?

**Response**
MLS data that we have downloaded from
http://avdc.gsfc.nasa.gov/pub/data/satellite/Aura/MLS/V04/L2GPOVP_Prof/O3/Summit/
are measured either between 5:28 and 6:26 UTC or between 14:11 and 15:10 UTC. These period concur roughly with sunrise and local solar noon at Summit, respectively. There is only one file per day provided by the Aura Validation Data Center, either from the earlier or later time range. There is no obvious difference in the timing between spring and fall. Of note, spectra used for the Global-Umkehr method were measured between 15:00 and 20:00 UTC.

MLS 1 data (i.e., MLS measurements taken before the period of Umkehr observations) and MLS 2 data (i.e., MLS data taken after this period) discussed in the manuscript were not filtered for time-of-day. The difference of MLS 2 and MLS 1 data showed virtually no bias (e.g., Figure 5l of manuscript), but there is some variability, which we attributed to several potential causes, including a real change in the ozone profile from one day to the next, variations in the horizontal distance between the locations of Summit and the MLS pixel from one day to the next, and random errors in MLS data.

Prompted by the referee's comment, we have now filtered and analyzed MLS data also by time-of-day. Specifically, we have calculated the difference in MLS profiles from one day to the next in dependence of the time-of-day when MLS 1 and MLS 2 measurements take place. Results are shown in the figure below.

[Figure]

Panel a is identical with Figure 5l of the manuscript and shows statistics of the difference of MLS 2 and MLS 1 data irrespective of time-of-day. It can be seen that there is virtually no bias between the two datasets.

Panel b is based on a subset of these data where both MLS 1 and MLS 2 are from the sunrise period. For Panel c, MLS 1 data were from the sunrise and MLS 2 data from the noon period. For Panel d, MLS 1 data were from the noon period and MLS 2 data from the sunrise period. Finally, Panel e shows the difference where both MLS 1 and MLS 2 data were filtered for the noon period.

A comparison of Panels b − e indicates that there is a bias between MLS 1 and MLS 2 measurements depending on whether data from the sunrise or noon periods are used. We did not consider this bias when submitting the original version of the manuscript. This bias agrees qualitatively with the difference of day − night profiles measured independently by several instruments at Mauna Loa, Hawaii (Parrish et al., 2014), and Bern, Switzerland (Studer et al., 2014). Specifically, data from Mauna Loa indicate 2-3% *higher* ozone concentrations during the day for pressure levels between 2 and 10 hPa (~ Umkehr layers 6 − 8) and 5-10% *lower* concentrations during the day for pressure levels between 0.5 and 1 hPa (Umkehr layer 10). At Bern, ozone concentrations between 3 and 10 hPa are highest in the afternoon, exceeding midnight concentrations by 3-5%. Above 2 hPa (~43 km), the pattern reverses with ozone concentrations being lower during the day than at night. Differences observed at Mauna Loa and Bern are by and large consistent with those shown in Panel c above and small differences between the three datasets may be explained by the different latitudes of Summit, Mauna Loa, and Bern.

As a result of these new findings, the manuscript was changed as follows:

- In Section 2, we added:
  "MLS measurements at Summit take place either between 5:28 and 6:26 UTC (period close to sunrise) or between 14:11 and 15:10 UTC (period close to local solar noon). There is only one data file per day in the NASA archive."

- The following was added to the Discussion:
  "Further analysis revealed that the difference between the MLS 1 and MLS 2 datasets depends also on the time when the daily MLS observation takes place. For example, when MLS 2 data are from the observation period close to local solar noon (14:11 to 15:10 UTC) and MLS 1 data are measured close to sunrise (5:28 to 6:26 UTC), MLS 2 data for Layers 7 − 9 are biased high by 3-6% relative to the MLS 1 dataset, while MLS 2 data for Layer 10 are biased low by 8%. This time-of-day dependency and its variation with altitude is by and large consistent with diurnal variations of the ozone profile measured by various instruments at Mauna Loa, Hawaii (Parrish et al., 2014), and by a microwave radiometer at Bern, Switzerland (Studer et al., 2014). This suggests that the

time-of-day effect observed at Summit is caused by actual diurnal changes of the ozone profile rather than potential time-dependent systematic errors in the MLS dataset. "

**The following references were added:**

Parrish, A., Boyd, I. S., Nedoluha, G. E., Bhartia, P. K., Frith, S. M., Kramarova, N. A., Connor, B. J., Bodeker, G. E., Froidevaux, L., Shiotani, M., and Sakazaki, T.: Diurnal variations of stratospheric ozone measured by ground-based microwave remote sensing at the Mauna Loa NDACC site: measurement validation and GEOSCCM model comparison, Atmos. Chem. Phys., 14(14), 7,255-7,272, doi:10.5194/acp-14-7255-2014, 2014.
Studer, S., Hocke, K., Schanz, A., Schmidt, H., and Kämpfer, N.: A climatology of the diurnal variations in stratospheric and mesospheric ozone over Bern, Switzerland, Atmos. Chem. Phys., 14(12), 5,905-5,919, doi:10.5194/acp-14-5905-2014, 2014.
* * *
**Referee comment**
Page 15, Fig. 4: It would be interesting to see if the change in season (thus, the vertical structure of the ozone profile) modifies the structure of the relative averaging kernels, especially, as fall and spring statistics are compared later on in Table 2.

**Response**
As stated in the text (e.g., P13, L18) the relative averaging kernels do not depend much on season. For the sake of simplicity, we therefore decided not to show the RAKs in Figure 3 and 4 at the time of the initial submission. Prompted by the referee's comment, we have now added the RAKs to the two figures.
* * *
**Referee comment**
Page 15, Fig 4: Also, why are the $\sigma_a = .4$ plots not shown in this figure? It would be interesting to see if the inversion agrees well in this case when it has more freedom due to a larger a priori covariance. If you have the results, they could also just be mentioned in the text.

**Response**
The effect of changing $\sigma_a$ from 0.1 to 0.4 is very similar for spring and fall profiles. For sake of brevity, we therefore did not include results for $\sigma_a = .4$, and still feel that this is the appropriate decision. This is also supported by the small difference in the statistics for spring and fall shown in Table 2.

We added the following to Sect. 3.1.3:
"The effect of changing $\sigma_a$ from 0.4 to 0.1 are similar for spring and fall profiles and results for $\sigma_a = 0.4$ were therefore omitted in Fig. 4."
* * *
**Referee comment**
At the beginning of the paper, you define Umkehr to refer to the standard zenith sky Umkehr technique and Global-Umkehr to refer to direct sun plus upper hemisphere. However, throughout the text and especially in the discussion you refer to Global-Umkehr as just Umkehr which is confusing. I suggest keeping the naming conventions consistent throughout the text.

**Response**
The naming convention was homogenized throughout the manuscript.
* * *
**Technical corrections**
Page 1, line 11: Substitute ultraviolet for UV.
Changed.

Page 1, line 18: The OMI acronym does not need to be included here as it is not
repeated in the abstract. It is redefined in the main text.
"OMI" deleted.

Page 2, line 4: Double closed bracket.
The second bracket belongs to "(e.g.," of the preceding line. No change.

Page 7, line 2: Is the AFGL acronym defined (Air Force Geophysics Laboratory)?
"Air Force Geophysics Laboratory" included in text.

Page 14, line 11: suggest changing identical to virtually identical as there is a small
difference of 1 DU as seen in Figure 4.
"virtually" included.

Page 15, line 7: Confusing sentence, suggest to change: ...they do not allow to assess
the Global-Umkehr technique comprehensively. to something like they do not allow the
comprehensive assessment of the Global-Umkehr technique.
Changed as suggested.

Page 5, line 10: Spaces seem to be present between all equations and symbols and
full stops, commas. This can be misleading in some instances. For example, Page 5,
line 10 may be interpreted as a dot product.
All equations will be reformatted by AMT before publication according to their guidelines. No
change to manuscript.

Table 1. There are spaces on either side of the endashes which are not consistent with
endash ranges throughout the text.
The entire manuscript, including the table, will be reformatted by AMT before publication
according to their guidelines. No change to manuscript.

Page 10, line 1: typo - MLS measure(s) thermal... - remove "s"
Grammar corrected.

Page 10, line 15: space after second open bracket.
Space removed.

Page 10, line 24: Suggest remove therefore or move to the start of the sentence -
Therefore,...
"Therefore" moved to front.

Page 17, line 15: should N be in parenthesis?
N enclosed in parentheses.

Page 19, line 8: Change Table 2 allows to assess retrievals... to something like Table 2 allows the assessment of retrievals...
Changed as suggested.

Page 19, lines 12 and 13: change to to between -6 % to 4 % and to between -5 % to 2 %
"between" inserted.

Page 19, line 18: remove is
"is" removed.

Page 19, lines 19 and 21: insert a space after the equals sign
Spaces inserted.
Page 19, line 19: Change to but it is consistent
"it" included.

Page 19, line 20: remove comma after standard
comma removed.

Page 20, line 8: Is (/2) meant to be there?
Yes. This is the way the SBUV instruments are identified by Miyagawa et al., 2014.

Page 20, line 23: change resembles to "resemble
Grammar corrected.

Page 22, line 8: change to ...have to be...
"be" included.

Page 22, line 24: change to ...2–3 % of those... (use an endash?)
Corrected as suggested.

---

## Author Comment (AC1)

**Response to comments of Referee #2**

We thank the referee for his or her thoughtful comments, which we have addressed as follows:

**Referee comment**
1. It is good to provide more details about how to derive a priori profiles from MLS and ozonesonde data. Are MLS data collocated with ozonesonde data around the Summit station? How MLS and ozonesonde data are merged as they cover different altitude ranges? Have other ozone profile climatologies such as McPeters et al. (2007) and McPeters and Labow (2012) been considered?

**Response**
MLS data are "overpass" data provided by the Aura Validation Data Center at https://avdc.gsfc.nasa.gov/pub/data/satellite/Aura/MLS/V04/L2GPOVP_Prof/O3/Summit/ as indicated in the manuscript. Data files provide the distance between the locations of Summit and the MLS profile. On average, the distance is 160 km. This was added to the manuscript.

As stated in the manuscript, "*A priori* state vectors $\mathbf{x}_a$ were constructed by combining balloon sonde profiles for altitudes below 10 km and profiles measured by the Microwave Limb Sounder (MLS) on NASA's Aura satellite for altitudes above 10 km […] Profiles for both seasons were constructed by calculating the median of a large number of sonde and MLS profiles measured during the two periods using data from the years 2004 to 2014." We believe that this description is sufficiently clear to indicate how the profiles were constructed.

We have not considered the ozone profile climatologies such as McPeters et al. (2007) and McPeters and Labow (2012) because we felt that *a priory* profiles constructed specifically for the location at Summit, separately for the spring and fall periods and using only data from the time period of relevance (2004 - 2014) are the most appropriate profiles.
* * *
**Referee comment**
2. Instead of using fixed a priori error of 0.1 and 0.4, you mentioned the use of altitude-dependent a priori errors in the discussion (P21 L18), which is likely more appropriate as the ozone variability is relatively small in most of the stratosphere, ~10% based on your analysis, but increases significantly in the lower stratosphere and upper troposphere to ~40%. You can modify Eq 4 to be more generic, allowing for altitude-dependent a priori errors: [Sa]mn = sigma_am^2 * [Xa]m * sigma_an^2 * [Xa]n * exp(- |m-n|/d)

**Response**
We agree that the altitude-dependence of the ozone variability could be considered when setting up the covariance matrix $\mathbf{S}_a$. For this reason, we mentioned this possibility in the Discussion as one of the options to optimize the Global-Umkehr method further. However, considering that the results obtained with $\sigma_a$ = 0.1 and 0.4 are fairly similar (see Table 2), we feel that the minor effect does not warrant a recalculation of all results.
* * *
**Referee comment**
3. P5, L8, one of the most important diagnostics is averaging kernels A, which is described in section 2.4. I suggested moving section 2.4 to in front of L8 as ds, is typically derived from A, as the trace of A. The diagonal elements of A are the ds at each layer.

**Response**
The contents of Sect. 2.4 were moved to the place suggested by the referee. We are aware that there are different methods to calculate $d_s$, all resulting in the same value for $d_s$. We describe the method suggested by Goering et al. (2005) because this is the method that was actually used in our calculations.
* * *
**Referee comment**
4. P6, Equation 8 is confusing. Looks like Dc(theta(t)) is not based on actual measurement, but based on the parameterization of clearly sky measurement as a function of SZA. You may change "Dc(theta(t)) is the measurement : " to "Dc(theta(t)) is the modeled photodiode measurement at time t that would be observed during clear skies, parameterized a function of SZA after filtering cloudy measurements." Also what criteria are used to filter cloudy measurements?

**Response**
The sentence was replaced with:
" $D_C(\theta(t))$ is the hypothetical clear-sky photodiode measurement at time $t$. The function was parameterized as a function of SZA using measurement of the photodiode obtained during clear skies. Clear-sky periods were determined based on temporal variability using the method described by Bernhard et al. (2008)."
* * *
**Referee comment**
5. It is better to switch sections "Retrieval method" and "Measurements" as the section of retrieval method depends on the description of measurements.

**Response**
We prefer to keep the original sequence of the manuscript, which starts with the fundamental equation of the optimal estimation approach (Gauss-Newton method) developed by Rodgers (2000). If we were to move the "Measurement" section to the front, we would also have to move the "Forward Model" section because the retrieval method also depends on the model. We feel that it is better to present the principle of the method first before discussing the various parameters that go into Eq. (1).
* * *
**Referee comment**
6. P6, L18-21, what is the main motivation of interpolating measurements to a common SZA grid that has 8 SZAs other than reducing the computation time. What is the typical number of spectra during the collection period (SZA change form 70 to 90)? Looks like it is much larger than 8, so interpolating it to 8 SZAs only while keeping the same measurement error can reduce the available information content and increase the measurement error. Have retrievals been conducted using the measurements at individual SZAs and compared with retrievals interpolated to 8 common SZAs?

**Response**
The interpolation of measurements to a common SZA grid is a common procedure for any Umkehr technique. For example, Petropavlovskikh et al. (2000) uses 14 fixed SZA between 60° and 90°. Using a common grid simplyfies calculations greatly. Note that our data always have to be interpolated because measurements at 310 and 337 nm are not performed at the same time and are therefore measured at slightly different SZAs. When developing our method, we also tried retrievals with up to eleven fixed SZAs. We found that there is virtually no benefit by increasing the number of SZAs beyond eight that would justify the greater computation time. This is not surprising because the "number of degrees of freedom for signal" $d_s$ is typically less than 3.1, suggesting that eight SZAs are more than sufficient to characterize the information content provided by the observations.

The number of spectra recorded during the observation time varied between 17 and 52, so there are enough spectra for accurate interpolations. Because we use approximating (smoothing) splines for interpolations, random errors are reduced, so even though retrievals are only based eight SZAs, we take advantage of the much larger number of spectra.

We have not conducted retrievals using the measurements at individual SZAs, but as noted above, results did not change significantly by using more than eight SZAs.

We added the following sentence to the manuscript:
"Tests indicated that retrieval results do not change significantly by adding measurements at additional SZAs."
* * *
**Referee comment**
7. P6, L30, why not using more recent ozone cross sections based on the activities of ACSO (Absorption Cross-Sections of Ozone) summarized in Orphal et al. (2016), which recommends that the BP data should not be used. Is this for consistency with the OMI TOC retrieval, which also used the BP data?

**Response**
Note that Referee 1 had a similar comment.
We are aware that more accurate ozone absorption cross sections than those published by Bass and Paur (1985) are now available and recommended. Nonetheless, we decided to use the Bass and Paur (1985) data because OMI total ozone data are based on B&P. Using a different cross section would have complicated the validation of our results with OMI data. We added the following to the manuscript:
"While more accurate ozone absorption cross sections are now available (Gorshelev et al., 2014; Orphal et al., 2016), we used Bass and Paur (1985) data to facilitate validation with OMI total ozone column measurements, which are also based on Bass and Paur (1985)."

**The following references were added:**

Gorshelev V., Serdyuchenko A., Weber M., Chehade W., and Burrows J.P.: High spectral resolution ozone absorption cross-sections–Part 1: Measurements, data analysis and comparison with previous measurements around 293K, Atmos. Meas. Tech., 7, 609-624, doi:10.5194/amt-7-609-2014, 2014.
Orphal J., Staehelin J., Tamminen J., Braathen G., De Backer M. R., Bais A., Balis D., Barbe A., Bhartia P. K., Birk M., and Burkholder J. B.: Absorption cross-sections of ozone in the

ultraviolet and visible spectral regions: Status report 2015, J. Mol. Spectrosc., 327, 105-121, doi:10.1016/j.jms.2016.07.007, 2016.

**Referee comment**

8. Are both SDISORT and MYSTIC RTMs based on scalar (rather than vector) radiative transfer models? If so, this is another source of forward model bias. What are the impacts of neglecting polarization (i.e., assuming scalar) on the calculated radiances? Just check if any such analysis has been done for either SIDOSRT and MYSTIC RTM.

**Response**

Yes, both SDISORT and MYSTIC were run in scalar mode. We agree with the referee that the MYSTIC results may be biases relative to the "truth" by neglecting polarization (i.e., by not using a vector model). Errors arising from neglecting polarization in radiative transfer calculations have been quantified by Lacis et al. (1998). The authors conclude that errors for radiances can be as large as 10% for a purely Rayleigh scattering atmosphere. However, most of these errors cancel when integrating over viewing angles to calculate global spectral irradiance. Fig. 1 below is reproduced from the top-right panel of Fig. 3 of Lacis et al. (1998) and shows error in irradiance for a Rayleigh atmosphere that result from the omission of polarization.

[Figure]

Fig. 1. (Scalar $-$ Vector)/Vector errors for a Rayleigh atmosphere adapted from Fig. 3 of Lacis et al. (1998). Rayleigh optical depths of relevance for Umkehr retrievals are indicated by grey horizontal lines.

Errors are plotted as a function of the cosine of the SZA (abscissa), $\mu_0$, and the Rayleigh optical depth, $\tau_R$, (ordinate). Relative errors range between $-1.3$ and 1.3%. For our Umkehr retrievals, only the difference in errors between measurements at 310 and 337 nm is relevant. For the altitude of Summit, $\tau_0(310)$ is 0.7 and $\tau_0(337)$ is 0.49. These optical depths are indicated by grey horizontal lines in the figure below. Further, SZAs range only between 90° and 70°,

corresponding to $0 \leq \mu_0 \leq 0.34$ (vertical broken line in the Fig. 1). It is apparent from Fig. 1 that the difference in irradiance errors for 310 and 337 nm is about 0.1% for all SZAs of relevance. According to Lacis et al. (1998), these errors are further reduced by a Lambertian surface, which is the case for Summit (snow surface with albedo of about 0.97).

Based on this analysis, we added the following to the Discussion:

"Finally, the MYSTIC Monte Carlo model, which was used to calculate the correction function $R(\theta)$ (see Fig. 1b), was run with a scalar radiative transfer solver, which did not take polarization into account. Lacis et al. (1998) calculated that modelling errors for irradiance resulting from the omission of polarization in these calculations can be as large as 1.3% for a Rayleigh atmosphere. However, errors for 310 and 337 nm (i.e., the wavelengths used in the Global-Umkehr method) agree to within 0.1%. We therefore conclude that the omission of polarization is not an import error source in our calculations."

**Reference:**
Lacis, A.A., Chowdhary, J., Mishchenko, M.I., and Cairns, B.: Modeling errors in diffuse-sky radiation: Vector vs. scalar treatment, Geophys. Res. Lett, 25(2), 135-138, 1998.
* * *
**Referee comment**
9. P10, L12, how is this threshold of 20 DU be determined?

**Response**
We assumed that changes in ozone occur linearly over time, so a 20 DU change over one day (as measured by OMI) translates to about 5 DU change over the course of the ~6 hours required for the Umkehr measurements. At Summit, the total ozone column varies between 250 and 350 DU during the spring period and between 320 and 480 DU during the fall period. So 5 DU make up about 1.6% of the column in spring and 1.3% in fall. Considering that day-to-day variations in the ozone profile occur mostly in the troposphere and lower stratosphere, relative ozone variations in these levels may exceed the percentages calculated for the column by a factor of about 2 to 3, resulting in relative variations of about 4% in these levels. Thus, by choosing a threshold of 20 DU, we ensure that variations in ozone over the course of the Umkehr observations are not a important uncertainty when comparing our results with MLS measurements. However, 20 DU is not a "magic number", and the criterion could be relaxed for operational processing.

We added the following to the manuscript:
"This criterion ensures that changes in the ozone profile remain below about 4% for all Umkehr layers."
* * *
**Referee comment**
10. P11, MLS measurements from consecutive days are used to quantify the temporal variation of ozone. It should be noted that the MLS measurements from consecutive days will be measured at different locations, maybe ~100 km apart. So some of the MLS1/2 difference is due to spatial variability. What is the average distance between MLS 1 and 2?

**Response**
We are aware of this problem and the following sentence had therefore been included in the Discussion of the original manuscript:

"However, a portion of the change in the MLS profile from one day to the next may be caused by the relatively poor horizontal resolution of MLS profiles of about 200 km. For example, some variability in the MLS overpass dataset can be attributed to the slightly different geolocation of two consecutive overpass profiles."

In response to the referee's comment, we have added the following:
"For example, the average horizontal distance between the locations of Summit and the MLS overpass is 160 km."

The number was calculated from the "Distance to the station" field provided in the MLS data files.

Of note, in addition to distance, the different viewing geometry of MLS 1 and 2 data and diurnal variations of the ozone profiles in the upper stratosphere are also important sources of variability. This point was raised by Referee #1. Please see our reply in response to the remark of Referee #1.
* * *
**Referee comment**
11. P12, L3-15, a lot of the description can be reduced as this has been described in the figure caption.

**Response**
The description was slightly reduced in length (see annotated manuscript).
* * *
**Referee comment**
12. P21 L20, you may consider using some recent cross sections as suggested in Orphal et al. (2016), and use meteorological data (e.g., temperature profiles) to account for the temperature dependence of the ozone absorption cross section. To reduce the impact of Ring effect, you may consider optimizing not only wavelengths, but also the magnitude of bandpass (currently 2 nm) used to degrade the spectral resolution. In addition, you can also mention the correction of forward model errors due to the neglect of polarization as commented earlier.

**Response**
The following was added:
"More current ozone absorption cross section data could be used (e.g., Orphal et al., 2016) than the Bass and Paur (1985) data implemented in this work and by OMI. If  temperature profile data are available, these could be utilized to account for the temperature dependence of the ozone absorption cross section."
and
"For example, by degrading the spectral resolution (currently set to 2 nm), the impact of the Ring effect could be reduced. Finally, the MYSTIC Monte Carlo model, which was used to calculate the correction function $R(\theta)$ (see Fig. 1b), was run with a scalar radiative transfer solver, which did not take polarization into account. Lacis et al. (1998) calculated that modelling errors for irradiance resulting from the omission of polarization in these calculations can be as large as 1.3% for a Rayleigh atmosphere. However, errors for 310 and 337 nm (i.e., the wavelengths used in the Global-Umkehr method) agree to within 0.1%. We therefore conclude that the omission of polarization is not an import error source in our calculations."
* * *
**Referee comment**

13. P22 L8, multiple scattering effect is also important for zenith sky measurements. You may say multiple scatterings effects become more important and the sphericity of the viewing geometry should be taken into account.

**Response**

Sentence changed to:

"Compared to the standard zenith-sky Umkehr method, multiple scattering effects become more important when exploiting global irradiance measurements, which also include contributions from photons received from directions close to the horizon. Therefore, the sphericity of the viewing geometry needs to be taken into account."
* * *
**Referee comment**

14. P24, L21, The poor sensitivity of the Umkehr method to ozone retrieval at layer 0 & 1 was mentioned here. Because only 2 wavelengths are used in the retrievals, measurements at other wavelengths especially the global irradiance spectrum can be used to improve the retrieval sensitivity in the first few layers as shown in Liu et al. (2005). You may add a few sentences about the possibility of exploring this for future studies.

**Response**

This comments seems to refer to P22, L21.

When we started working on the Global-Umkehr method, we were aware that tropospheric ozone profiles can be retrieved from off-axis radiance measurement by using more than two wavelengths in the UV. In addition to Liu et al. (2005), this method has for example also been used by Tzortziou et al. (2008). When developing our method we therefore explored retrievals with additional UV wavelengths, specifically combinations of E(305)/E(337); E(310)/E(337); E(325)/E(337), see P3, L5 of manuscript. We were hopeful that the tropospheric resolution of the Global-Umkehr method could be improved by including this additional spectral information. Unfortunately, adding these additional wavelength pairs did not lead to a significant improvement. We are therefore not hopeful that tropospheric ozone profiles can be retrieve from global spectral irradiance spectra without adding additional viewing angles, which is not possible for our instrument. We therefore do not think that it is appropriate to raise hopes. No change to manuscript.
* * *
**Technical comments**

1. P2, L20, change "a.s.l" to "a.s.l."
Period added.

2. The section of "1 Method" should be "2 Method" and "1.1 Retrieval method" should be "2.1 Retrieval method"
Corrected.

3. P2, L25, change "depends" to "depend"
Changed. (Typo was on P4, L25).

4. P5, L17, "and is part of… :"

No change because the sentence is an enumeration with the phrase in question being the second of three phrases.

5. P5, L27, change "wavelengths shifts" to "wavelength shifts"
Extra "s" removed.

6. P6, L17, this sentence can be grouped to the above paragraph.
Paragraph break removed.
7. P6, L28, change "result are" to "results are"
"s" inserted.

8. P7, L6, change "reference" to "references"
"s" inserted.

9. P7, L24, change "considered" to "considered as"
"as" inserted.

10. P15, L18, add "," before "interquartile"
Comma inserted.

11. P19, L28, change "decreased" to "has decreased" or "decreases"
"has" inserted (but on P18, L28)

12. P19, L18, change "is varies" to "varies"
"is" deleted.

13. P19 L26ïıjˇN change to "compared"
"McElroy and Kerr (1995) compare Umkehr profiles" changed to "McElroy and Kerr (1995) compared Umkehr profiles"

14. P20, L1, L2, L5, L6, change to "compared", "found", "concluded", "compared"
Changed to past tense as suggested.

15. P23 L10, change to "on a weekly basis"
"a" inserted.

16. P24, L17, add "," after "Phys."
Comma inserted.

17. P24, L18, add "," after "Res."
Comma inserted.

18. P26, last line, use normal font for the journal title.
Italic formatting removed.